# The Rich and the Simple:
# On the Implicit Bias of Adam and SGD

**Bhavya Vasudeva**   **Jung Whan Lee**   **Vatsal Sharan**   **Mahdi Soltanolkotabi**
Department of Computer Science
University of Southern California
{bvasudev,jlee7870,vsharan,soltanol}@usc.edu

## Abstract

Adam is the de facto optimization algorithm for several deep learning applications, but an understanding of its implicit bias and how it differs from other algorithms, particularly standard first-order methods such as (stochastic) gradient descent (GD), remains limited. In practice, neural networks (NNs) trained with SGD are known to exhibit simplicity bias — a tendency to find simple solutions. In contrast, we show that Adam is more resistant to such simplicity bias. First, we investigate the differences in the implicit biases of Adam and GD when training two-layer ReLU NNs on a binary classification task with Gaussian data. We find that GD exhibits a simplicity bias, resulting in a linear decision boundary with a suboptimal margin, whereas Adam leads to much richer and more diverse features, producing a nonlinear boundary that is closer to the Bayes' optimal predictor. This richer decision boundary also allows Adam to achieve higher test accuracy both in-distribution and under certain distribution shifts. We theoretically prove these results by analyzing the population gradients. Next, to corroborate our theoretical findings, we present extensive empirical results showing that this property of Adam leads to superior generalization across various datasets with spurious correlations where NNs trained with SGD are known to show simplicity bias and do not generalize well under certain distributional shifts.

## 1   Introduction

Adaptive optimization algorithms, particularly Adam [1], have become ubiquitous in training deep neural networks (NNs) due to their faster convergence rates and better performance, particularly on large language models (LLMs), as compared to (stochastic) gradient descent (SGD) [2]. Despite its widespread use, the theoretical understanding of how Adam works and when/why it outperforms (S)GD remains limited.

Modern NNs are heavily overparameterized and thus the training landscape has numerous global optima. As a result, different training algorithms may exhibit preferences or biases towards different global optima a.k.a. *implicit bias*. There is extensive prior work on the implicit bias of GD [3, 4, 5, 6], for both linear and nonlinear models (see Section 5 for a detailed discussion of related work). However, there is limited work investigating the implicit bias of Adam. Recently, Zhang et al. [7] showed that for linear logistic regression with separable data, Adam iterates directionally converge to the minimum $\ell_\infty$-norm solution, in contrast to GD which converges to minimum $\ell_2$-norm [3]. This difference between the implicit bias of Adam vs GD in simple linear settings motivates the central question of this paper:

*What is the implicit bias of Adam for nonlinear models such as NNs, and how does it differ from the implicit bias of (S)GD?*

Given the popularity of Adam, surprisingly little is known about the implicit bias of Adam for training NNs. A notable exception is the recent work Tsilivis et al. [8], which characterized the

39th Conference on Neural Information Processing Systems (NeurIPS 2025).

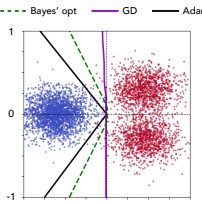

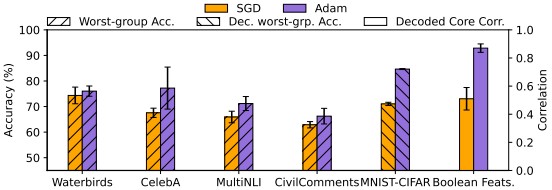

Figure 1: Illustration of the synthetic dataset considered in this work, and comparison of the Bayes' optimal predictor with the decision boundaries of two-layer NNs trained with Adam and GD.

Figure 2: Training with Adam leads to better performance across different test set metrics on six benchmark datasets with spurious correlations, as compared to SGD. See Section 4 for details.

late-stage implicit bias of a family of steepest descent algorithms, on homogeneous NNs, in terms of maximizing an algorithm-dependent geometric margin ($\ell_\infty$ for signGD or Adam without momentum) and convergence to KKT points. However, the stationary points of the respective margin-maximization may not be unique. Additionally, this characterization does not relate the implicit bias of the algorithm with properties of the learned solution, such as the type of features learned, or the complexity of the decision boundary, which in turn impact its generalization. In particular, in many applications, NNs trained with SGD are known to exhibit a *simplicity bias*, that is, they learn simple solutions [9], which can lead to suboptimal generalization. For example, SGD-trained two-layer NNs rely on low-dimensional projections of the data to make predictions [10]. This simplicity bias can be particularly detrimental in the presence of spurious features where it is *simpler* for NNs trained with SGD to utilize them to achieve zero training error.* This leads to the question: Does training with Adam lead to solutions that are resistant to this simplicity bias?

In this paper, we answer these questions in two ways. *Theoretically*, we show that two-layer ReLU NNs trained on a Gaussian mixture data setting (see Fig. 1) with SGD exhibit simplicity bias while training with Adam leads to richer feature learning. *Empirically*, we demonstrate that training with Adam can lead to better performance as compared to SGD on various benchmark datasets with spurious features (see Fig. 2). Our main contributions are as follows.

- We identify a simple yet informative setting with mixture of Gaussians where GD and Adam exhibit different implicit biases (see Fig. 1). The Bayes' optimal predictor in this setting is nonlinear (piecewise linear), and *we show — both theoretically and empirically — that while GD exhibits a simplicity bias resulting in a linear predictor, Adam encourages reliance on richer features leading to a nonlinear decision boundary*, which is closer to the Bayes' optimal predictor. We theoretically prove this difference in the implicit bias by analyzing the population gradients and updates of GD and Adam without momentum (signGD). We also show that this leads to better test accuracy in distribution as well as across some distribution shifts. Additionally, to theoretically understand the behaviour of Adam with momentum, we analyze a simpler setting where the variance of the Gaussians approaches 0, in the infinite width limit. We show that the decision boundaries learned with signGD and Adam are more nonlinear (and closer to the Bayes' optimal predictor) than the one learned with GD.

- *We conduct extensive experiments on various datasets and show that Adam leads to richer features that are more robust compared to simpler features learned via SGD, allowing Adam to achieve higher test accuracy both in distribution and under certain distribution shifts*. First, we consider an MNIST-based task with a colored patch as a spurious feature and show that compared to SGD, training with Adam leads to a more nonlinear decision boundary, larger margins overall, and has better generalization on a test set with flipped correlation. Next, we show that Adam achieves better worst-group accuracy on four benchmark datasets (Waterbirds, CelebA, MultiNLI, and CivilComments) for subgroup robustness, and better decoded worst-group accuracy on the Dominoes or MNIST-CIFAR dataset [9], with images from CIFAR and MNIST classes as the complex/core and simple/spurious features, respectively. Finally, we study the Boolean features

---

*We remark that simplicity bias may not always be detrimental; for instance, it can be beneficial for in-distribution generalization (see Appendix C for further discussion). Our focus is characterizing and contrasting the implicit bias of Adam vs (S)GD in terms of rich vs simple feature learning, not advocating for one to always be better.

dataset proposed in Qiu et al. [11], and show that training with Adam leads to better core feature learning as compared to SGD.

## 2 Setup

We consider a two-layer homogeneous neural network with fixed final layer and ReLU activation, defined as $f(\boldsymbol{W}; \boldsymbol{x}) := \boldsymbol{a}^\top \sigma(\boldsymbol{W}\boldsymbol{x})$, where $\boldsymbol{x} \in \mathbb{R}^d$ denotes the input, $\boldsymbol{W} \in \mathbb{R}^{m \times d}$ denotes the trainable parameters, $\boldsymbol{a} \in \{\pm 1\}^m$ are the final layer weights, and $\sigma(\cdot) := \max(0, \cdot)$ is the ReLU activation. Let $S := \{(\boldsymbol{x}_i, y_i)\}_{i=1}^n$ denote the set of train samples, where the label $y \in \{-1, 1\}$. The model is trained to minimize the empirical risk $\widehat{L}(\boldsymbol{W}) := \frac{1}{n} \sum_{i=1}^n \ell(-y_i f(\boldsymbol{W}; \boldsymbol{x}_i))$, where $\ell$ denotes a decreasing loss function. We consider two loss functions, namely logistic loss, where $\ell(z) := \log(1 + \exp(z))$, and correlation or linear loss, where $\ell(z) := z$, for $z \in \mathbb{R}$. We focus on the following two update rules.

**Gradient Descent.** The updates for GD with step-size $\eta > 0$ at iteration $t \geq 0$ are written as $\boldsymbol{W}_{t+1} = \boldsymbol{W}_t - \eta \boldsymbol{G}_t$, where $\boldsymbol{G}_t := \nabla_{\boldsymbol{W}} \widehat{L}(\boldsymbol{W}_t)$, each row of which is written as:

$$\frac{-1}{n} \sum_{i=1}^n \ell'_{i,t} y_i \nabla_{\boldsymbol{w}_j} f(\boldsymbol{W}_t; \boldsymbol{x}_i) = \frac{-1}{n} \sum_{i=1}^n \ell'_{i,t} a_j \sigma'(\boldsymbol{w}_{j,t}^\top \boldsymbol{x}_i)(y_i \boldsymbol{x}_i),$$

where $\ell'_{i,t}$ denotes $\ell'(-y_i f(\boldsymbol{W}_t, \boldsymbol{x}_i))$ for convenience, and $\sigma'(z) := \mathbb{1}[z \geq 0]$, for $z \in \mathbb{R}$.

**Adam.** The update rule for the Adam optimizer [1] is as follows:

$$\boldsymbol{W}_{t+1} = \boldsymbol{W}_t - \eta \hat{\boldsymbol{M}}_t \odot (\hat{\boldsymbol{V}}_t + \epsilon \mathbf{1}\mathbf{1}^\top)^{\circ - 1/2},$$

where $\hat{\boldsymbol{M}}_t = \frac{\boldsymbol{M}_{t+1}}{1 - \beta_1^{t+1}} = \frac{1}{1 - \beta_1^{t+1}} \left( \beta_1 \boldsymbol{M}_t + (1 - \beta_1) \boldsymbol{G}_t \right)$ is the bias-corrected first-moment estimate,

and $\hat{\boldsymbol{V}}_t = \frac{\boldsymbol{V}_{t+1}}{1 - \beta_2^{t+1}} = \frac{1}{1 - \beta_2^{t+1}} \left( \beta_2 \boldsymbol{V}_t + (1 - \beta_2) \boldsymbol{G}_t \odot \boldsymbol{G}_t \right)$ is the bias-corrected second (raw) moment estimate. $\epsilon$ is the numerical precision parameter, which is set as $0$ for the theoretical results. Also, $\odot$ and $(\cdot)^\circ$ denote the Hadamard product and power, respectively, and $\boldsymbol{M}_0$ and $\boldsymbol{V}_0$ are initialized as zeroes. Note that we can write

$$\hat{\boldsymbol{M}}_t = \frac{\sum_{\tau=0}^t \beta_1^\tau \boldsymbol{G}_{t-\tau}}{\sum_{\tau=0}^t \beta_1^\tau} \quad \text{and} \quad \hat{\boldsymbol{V}}_t = \frac{\sum_{\tau=0}^t \beta_2^\tau \boldsymbol{G}_{t-\tau} \odot \boldsymbol{G}_{t-\tau}}{\sum_{\tau=0}^t \beta_2^\tau}.$$

At each optimization step, the descent direction is different from the gradient direction because of the entry-wise division with the second (raw) moment. Further, the first update step exactly matches the update of signGD, which uses the sign of the gradient $\text{sign}(\boldsymbol{G}_t)$ instead of directly using the gradient $\boldsymbol{G}_t$ for the update. This is because $\hat{\boldsymbol{M}}_0 = \boldsymbol{G}_0$ and $\hat{\boldsymbol{V}}_0 = \boldsymbol{G}_0 \odot \boldsymbol{G}_0$, and hence $(\hat{\boldsymbol{M}}_0 \odot \hat{\boldsymbol{V}}_0^{\circ - 1/2})_{i,j} = \frac{(G_0)_{i,j}}{|(G_0)_{i,j}|} = \text{sign}((G_0)_{i,j})$. Similarly, when the parameters $\beta_1$ and $\beta_2$ are set as $0$, the Adam updates are the same as signGD for every $t \geq 0$.

**Dataset.** Our synthetic dataset is designed to investigate the impact of feature diversity on the implicit biases of GD and Adam in NN training. It models two classes with differing feature distributions to emulate real-world scenarios where feature complexity may vary between classes. See Fig. 1 for an illustration of the dataset. Concretely, each sample $(\boldsymbol{x}, y)$ is generated as follows:

$$y \sim \text{Unif}(\{\pm 1\}), \quad \epsilon \sim \text{Unif}(\{\pm 1\}) \tag{1}$$

$$x_1 \sim \mathcal{N}\left( \frac{\mu_1 - \mu_3}{2} + y \frac{\mu_1 + \mu_3}{2}, \sigma_x^2 \right), \quad x_2 \sim \mathcal{N}\left( \epsilon \left( \frac{y+1}{2} \right) \mu_2, \sigma_y^2 \right), \quad x_j \sim \mathcal{N}(0, \sigma_z^2), \forall j \in \{3, \dots, d\}.$$

The first two dimensions contain information about the label while the rest are noisy. Our dataset construction is inspired by the synthetic "slabs" dataset introduced by Shah et al. [9]. While their approach utilizes slab features to represent non-linearly separable components, we consider Gaussian features instead. This modification enhances the realism of the synthetic data and facilitates a more nuanced analysis of the NN training dynamics.

We first write the Bayes' optimal predictor for this dataset when using only the signal dimensions as follows.

**Proposition 1** (Bayes' Optimal Predictor). *The optimal predictor for the data in Eq.* (1) *with* $d = 2$ *is:*

$$(\mu_1 + \mu_3)x_1 + \frac{\sigma_x^2}{\sigma_y^2} \mu_2 x_2 = \frac{\mu_1^2 - \mu_3^2}{2} + \frac{\mu_2^2 \sigma_x^2}{2\sigma_y^2} - \sigma_x^2 \log\left( 0.5 \left( 1 + \exp\left( -\frac{2\mu_2 x_2}{\sigma_y^2} \right) \right) \right).$$

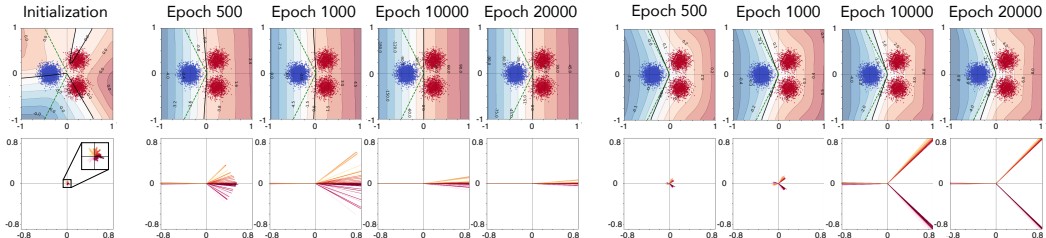

Figure 3: Evolution of the decision boundary (top row) and the neurons (bottom row) over time, for GD (left) and Adam (right) with learning rates $0.1$ and $10^{-4}$ over $20\,000$ epochs of training a width 100 NN with small initialization (the neurons are colored based on the quadrant they were initialized in) using population gradients (the samples are plotted for illustration purposes) on the Gaussian data setting (Eq. (1)) with $\mu = 0.3, \omega = 2, \sigma = 0.1$. GD leads to a linear decision boundary, with neurons mostly aligned with the directions $[\pm 1, 0]^\top$, while Adam (with $\beta_1 = \beta_2 = 0.9999$) leads to a non-linear decision boundary, with neurons aligned with three main directions $[-1, 0]^\top, [1, 1]^\top, [1, -1]^\top$, which is closer to the Bayes' optimal predictor.

Since we consider homogeneous NNs (no bias parameter), we make the following assumption on the data generating process to make the setting realizable, *i.e.*, ensure that the Bayes' optimal predictor passes through the origin.

**Assumption 1** (Realizability). *Let* $\mu := \mu_2$, $\kappa := \frac{\sigma_x^2}{\sigma_y^2}$, $\omega := \frac{\mu_1 + \mu_3}{\kappa\mu} \geq 1$. *For realizability,* $\mu_1 = \frac{\mu}{2}\left(\kappa\omega - \frac{1}{\omega}\right)$ *and* $\mu_3 = \frac{\mu}{2}\left(\kappa\omega + \frac{1}{\omega}\right)$.

Here, $\kappa$ denotes the degree of anisotropy of the clusters, and $\pm\frac{1}{\omega}$ corresponds to the slopes of the two linear components of the optimal predictor.

## 3 Theoretical Results

In this section, we aim to theoretically analyze GD and Adam and the differences in the learned solution arising from the update rules. For clarity of exposition we focus on a simple setting. *Specifically, in this section, we consider the infinite sample limit, fixed outer layer weights, $d = 2$, and training with correlation or linear loss.* As we will see later in Section 4, these algorithms learn different solutions even when these assumptions are relaxed.

### 3.1 Gaussian Data

We first obtain a closed-form of the population gradient for the Gaussian cluster data as follows. The proof is included in Appendix A.

**Proposition 2** (Population Gradient). *Consider the data in Eq. (1) with* $\sigma_x = \sigma_y = \sigma_z = \sigma$. *When using correlation loss, the population gradient for neuron $\boldsymbol{w}$ is written as:*

$$\nabla_{\boldsymbol{w}}\widehat{L}(\boldsymbol{W}) = -a\mathbb{E}_{(\boldsymbol{x},y)\sim\mathcal{D}}\big[\mathbb{1}\big[\boldsymbol{w}^\top\boldsymbol{x} \geq 0\big]y\boldsymbol{x}\big]$$

$$= -\frac{a\sigma}{4}\Big(\Phi(\lambda\bar{\boldsymbol{\mu}}_+^\top\bar{\boldsymbol{w}})\lambda\bar{\boldsymbol{\mu}}_+ + \Phi(\lambda\bar{\boldsymbol{\mu}}_-^\top\bar{\boldsymbol{w}})\lambda\bar{\boldsymbol{\mu}}_- - 2\Phi(\lambda\bar{\boldsymbol{\mu}}_0^\top\bar{\boldsymbol{w}})\lambda\bar{\boldsymbol{\mu}}_0 + \big(\phi(\lambda\bar{\boldsymbol{\mu}}_+^\top\bar{\boldsymbol{w}}) + \phi(\lambda\bar{\boldsymbol{\mu}}_-^\top\bar{\boldsymbol{w}}) - 2\phi(\lambda\bar{\boldsymbol{\mu}}_0^\top\bar{\boldsymbol{w}})\big)\bar{\boldsymbol{w}}\Big),$$

*where* $\lambda := \frac{\mu}{\sigma}\frac{\omega^2+1}{2\omega}$ $\bar{\boldsymbol{\mu}}_+ := \left[\frac{\omega^2-1}{\omega^2+1}, \frac{2\omega}{\omega^2+1}, 0, \ldots, 0\right]^\top$, $\bar{\boldsymbol{\mu}}_- := \left[\frac{\omega^2-1}{\omega^2+1}, -\frac{2\omega}{\omega^2+1}, 0, \ldots, 0\right]^\top$, $\bar{\boldsymbol{\mu}}_0 := [-1, 0, 0, \ldots, 0]^\top$, *and $\phi$ and $\Phi$ denote the normal PDF and CDF, respectively.*

Here, $\bar{\boldsymbol{\mu}}_+, \bar{\boldsymbol{\mu}}_-$ denote the (normalized) mean vectors of the clusters with label 1 and $\bar{\boldsymbol{\mu}}_0$ denotes the mean of the cluster with label $-1$.

Before theoretically analyzing the training dynamics of GD and Adam, we compare them empirically. Fig. 3 shows the evolution of the decision boundary and the neurons for two-layer NNs trained with GD and Adam using the population gradients in Proposition 2, as a function of the training epochs. We observe that for GD (left), all neurons align in direction $[\pm 1, 0]^\top$, and the learned decision boundary is linear, *i.e.*, the model only relies on the first dimension to make predictions. On the other hand, for Adam (right), neurons converge along three different directions, $[-1, 0]^\top, \frac{1}{\sqrt{2}}[1, 1]^\top, \frac{1}{\sqrt{2}}[1, -1]^\top$, and the learned decision boundary is piece-wise linear, *i.e.*, the model uses both the signal dimensions.

We will now prove these results for the two optimizers in Theorem 1 and Theorem 2. First, we leverage the gradient expression in Proposition 2 to show that gradient descent with infinitesimal step size, *i.e.*, gradient flow (GF) exhibits simplicity bias and learns a linear predictor.

**Theorem 1.** *(Informal) Consider the data in Eq. (1), neurons initialized such that $a_k = \pm 1$ with probability $0.5$, $d = 2$, $\sigma_x = \sigma_y = \sigma$, and $\omega > c_1$ and $\frac{\mu}{\sigma} > c_2$, where $c_1, c_2$ are constants. Let $\boldsymbol{w}_{k,\infty} := \lim_{t\to\infty} \frac{\boldsymbol{w}_{k,t}}{t}$ and $\bar{\boldsymbol{w}}_{k,\infty} := \frac{\boldsymbol{w}_{k,\infty}}{\|\boldsymbol{w}_{k,\infty}\|}$, for $k \in [m]$. Then, the solution learned by gradient flow is: $\bar{\boldsymbol{w}}_{k,\infty} = a_k[1,0]^\top$.*

The proof is included in Appendix A. The main step in the proof is using the relation $\cos\theta_{k,t} = \bar{\boldsymbol{w}}_{k,t}^\top \bar{\boldsymbol{w}}^*$, where $\bar{\boldsymbol{w}}^* := [1,0]^\top$, and showing that

$$a_k \frac{d\cos\theta_{k,t}}{dt} = a_k \dot{\bar{\boldsymbol{w}}}_{k,t}^\top \bar{\boldsymbol{w}}^* = a_k \frac{\dot{\boldsymbol{w}}_{k,t}^\top}{\|\boldsymbol{w}_{k,t}\|}(I - \bar{\boldsymbol{w}}_{k,t}\bar{\boldsymbol{w}}_{k,t}^\top)\bar{\boldsymbol{w}}^* > C\frac{(\sin\theta_{k,t})^2}{\|\boldsymbol{w}_{k,t}\|},$$

where $C > 0$ is a constant, for every $t > 0$. To show this, we use the fact that $\dot{\boldsymbol{w}}_{k,t} = -\nabla_{\boldsymbol{w}_{k,t}}\widehat{L}(\boldsymbol{W}_t)$ for gradient flow, and use the population gradient from Proposition 2. Further, using Proposition 2 and showing that $\|\boldsymbol{w}_{k,t}\| \le c(t+1)$ for some constant $c > 0$, we prove by contradiction that $\sin\theta_{k,t} \to 0$ as $t \to \infty$, and hence $\cos\theta_{k,t} \to 1$.

Next, we analyze Adam with $\beta_1 = \beta_2 = 0$ (signGD), and show that it learns both features resulting in a nonlinear predictor.

**Theorem 2.** *(Informal) Consider the data in Eq. (1), neurons initialized such that $a_k = \pm 1$ with probability $0.5$, small initialization scale, $d = 2$, $\sigma_x = \sigma_y = \sigma$, $\omega > c$ and $c_1 \le \frac{\mu}{\sigma} \le c_2$, where $c, c_1, c_2$ are constants. Let $\boldsymbol{w}_{k,\infty} := \lim_{t\to\infty} \frac{\boldsymbol{w}_{k,t}}{t}$ and $\bar{\boldsymbol{w}}_{k,\infty} := \frac{\boldsymbol{w}_{k,\infty}}{\|\boldsymbol{w}_{k,\infty}\|}$, for $k \in [m]$. Let $\theta_0$ denote the direction of $\boldsymbol{w}_{k,0}$. Then, the solution learned by signGD is:*

$$\bar{\boldsymbol{w}}_{k,\infty} = \begin{cases} \frac{1}{\sqrt{2}}[1,1]^\top & a_k > 0, \sin\theta_{k,0} > 0, \\ \frac{1}{\sqrt{2}}[1,-1]^\top & a_k > 0, \sin\theta_{k,0} < 0, \\ [1,0]^\top & a_k > 0, \sin\theta_{k,0} = 0, \\ [-1,0]^\top & a_k < 0. \end{cases}$$

The proof is included in Appendix A. At a high level, we leverage Proposition 2 to show that at each $t > 0$, the gradient update in each of the two dimensions is nonzero and signGD updates are in the direction $[\texttt{sign}(a_k), \texttt{sign}(a_k \sin\theta_{k,t})]^\top$. Notably, when $a_k > 0$, neurons are either in $\frac{1}{\sqrt{2}}[1,1]^\top$ or in $\frac{1}{\sqrt{2}}[1,-1]^\top$ direction at each iteration. However, when $a_k < 0$, the neurons point to $\frac{1}{\sqrt{2}}[-1,\pm 1]^\top$ and $\frac{1}{\sqrt{2}}[-1,\mp 1]^\top$ in alternating iterations, leading to convergence in the $[-1,0]^\top$ direction.

These results characterize the direction in which each neuron converges asymptotically. For GF, all neurons are in the same direction, with exactly half the neurons in $[1,0]^\top$ and $[-1,0]^\top$ directions, which leads to a linear predictor. In contrast, for signGD, there is a fraction of neurons aligned in the directions $\frac{1}{\sqrt{2}}[1,1]^\top$ and $\frac{1}{\sqrt{2}}[1,-1]^\top$ which leads to a piece-wise linear decision boundary. These results explain the behaviour we observed in Fig. 3. We note that while our theoretical result considers GF, the continuous time version of GD, it is still predictive of the behaviour we observe for discrete-time GD with small step-size. We also observe a similar behaviour for Adam vs GD in Fig. 1, where we consider finite samples (see Section 4 for details).

Analyzing this setting allows us to conceptually understand how Adam (without momentum) operates and leads to rich feature learning, while GD exhibits simplicity bias. Importantly, we make no assumptions regarding the initialization direction of the neural network parameters, ensuring that any differences observed between Adam and GD arise solely from the inherent characteristics of the optimization algorithms themselves. Since we analyze the population setting, these results are in the under-parameterized regime and don't require any lower bound on the network width $m$ to ensure overparameterization.

Next, we show that under some conditions on the distribution parameters, the piece-wise linear predictor learned by Adam (without momentum) obtains a strictly lower test error than the linear predictor learned by GD. For simplicity, we assume that $m \to \infty$ for the following result, so that we can write the predictor learned by Adam in a piece-wise linear form which is symmetric (with respect to the first dimension), using $p(\sin\theta_{k,0} > 0) = p(\sin\theta_{k,0} < 0) = 0.5$. However, even with finite $m$, these probabilities concentrate well and we can expect the following to still hold.

**Theorem 3.** *(Informal) Consider the data in Eq. (1) with $d = 2$ and $\omega = \Theta(1)$, $\kappa = \frac{\sigma_x^2}{\sigma_y^2} \in \left[\frac{1}{\omega^2}, 1\right]$, $\frac{\mu}{\sigma_y} \geq c\sqrt{\kappa}\omega$, where $c$ is a constant. Consider two predictors,*

$$\text{Linear: } \hat{y} = sign(x_1),$$
$$\text{Piece-wise Linear: } \hat{y}' = \begin{cases} sign(3x_1 + x_2) & \text{if } x_2 \geq 0, \\ sign(3x_1 - x_2) & \text{if } x_2 < 0. \end{cases}$$

*Then, it holds that $\mathbb{E}(\hat{y}' \neq y) - \mathbb{E}(\hat{y} \neq y) < 0$.*

The proof is included in Appendix A. Note that we consider isotropic distributions ($\kappa = 1$) for Theorems 1 and 2, whereas the above result on the test error applies to $\kappa \in \left[\frac{1}{\omega^2}, 1\right]$. This shows that training with Adam can provably lead to better test accuracy both in-distribution and across certain distribution shifts.

In the next section, we consider a simplified setting to investigate the effect of setting $\beta_1, \beta_2 \approx 1$ for Adam, which is closer to the setting used in practice, where $\beta_1 = 0.9, \beta_2 = 0.999$.

### 3.2 Toy Data Setting

We consider a simple yet informative setting where $d = 2$ and $\sigma_x = \sigma_y = 0$, which we refer to as the toy data setting. Specifically, the samples are generated as follows:

$$y \sim \text{Unif}(\{\pm 1\}), \quad \epsilon \sim \text{Unif}(\{\pm 1\}), \quad x_1 = \frac{\mu}{2}\left(y\omega - \frac{1}{\omega}\right), \quad x_2 = \epsilon\frac{y+1}{2}\mu. \tag{2}$$

This setting allows us to characterize the full trajectory of each neuron for the three algorithms, namely GD, signGD, and Adam ($\beta_1 = \beta_2 \approx 1$). We now state our main result.

**Theorem 4.** *(Informal) Consider the toy data in Eq. (2), neurons initialized at a small scale, and $c_1 < \omega < c_2$, where $c_1, c_2$ are constants. Let $\boldsymbol{w}_{k,\infty} := \lim_{t\to\infty}\frac{\boldsymbol{w}_{k,t}}{t}$ and $\bar{\boldsymbol{w}}_{k,\infty} := \frac{\boldsymbol{w}_{k,\infty}}{\|\boldsymbol{w}_{k,\infty}\|}$, for $k \in [m]$ and $p := \frac{\tan^{-1}\frac{\omega^2-1}{2\omega}}{\pi}$. Then, for $m \to \infty$, the solutions learned by GD, signGD, and Adam are shown in Table 1, where $s$ is a constant $\in [0.72, 1]$, the probabilities are over the neurons, and the sign of the first element of $\boldsymbol{w}_{k,\infty}$ is the same as $sign(a_k)$.*

| | GD | | Adam ($\beta_1 = \beta_2 = 0$) or signGD | | Adam ($\beta_1 = \beta_2 \approx 1$) | |
|---|---|---|---|---|---|---|
| $\bar{\mathbf{w}}_{k,\infty} =$ | $\begin{cases} [1,0]^\top \\ [-1,0]^\top \\ \frac{1}{\omega^2+1}\left[\omega^2-1, 2\omega\right]^\top \\ \frac{1}{\omega^2+1}\left[\omega^2-1, -2\omega\right]^\top \end{cases}$ | $\begin{matrix} \text{w.p. } \frac{1}{4}+\frac{p}{2} \\ \text{w.p. } \frac{1}{2} \\ \text{w.p. } \frac{1}{8}-\frac{p}{4} \\ \text{w.p. } \frac{1}{8}-\frac{p}{4} \end{matrix}$ | $\begin{cases} [1,0]^\top \\ [-1,0]^\top \\ \frac{1}{\sqrt{2}}[1,1]^\top \\ \frac{1}{\sqrt{2}}[1,-1]^\top \end{cases}$ | $\begin{matrix} \text{w.p. } p \\ \text{w.p. } \frac{1}{2} \\ \text{w.p. } \frac{1}{4}-\frac{p}{2} \\ \text{w.p. } \frac{1}{4}-\frac{p}{2} \end{matrix}$ | $\begin{cases} [1,0]^\top \\ [-1,0]^\top \\ \frac{1}{\sqrt{2}}[1,1]^\top \\ \frac{1}{\sqrt{2}}[1,-1]^\top \\ \frac{1}{\sqrt{s^2+1}}[s,1]^\top \\ \frac{1}{\sqrt{s^2+1}}[s,-1]^\top \end{cases}$ | $\begin{matrix} \text{w.p. } p \\ \text{w.p. } \frac{1}{2} \\ \text{w.p. } \frac{1}{8}-\frac{p}{4} \\ \text{w.p. } \frac{1}{8}-\frac{p}{4} \\ \text{w.p. } \frac{1}{8}-\frac{p}{4} \\ \text{w.p. } \frac{1}{8}-\frac{p}{4} \end{matrix}$ |

Table 1: Solutions learned by different algorithms on the toy dataset (see Theorem 4).

Note that $p < 0.5$, which implies (from Table 1) that the fraction of neurons in the direction $[1,0]^\top$ is larger for GD as compared to Adam. Consequently, the decision boundary learned by Adam is more non-linear. The proof mainly relies on analyzing the updates of each algorithm, so we defer it to Appendix A. Fig. 4 shows the decision boundaries learned by the three algorithms, as mentioned in Table 1. The predictor learned by Adam is more non-linear and closer to the Bayes' optimal predictor.

We note that the difference between the predictors learned by GD and Adam is more significant in the Gaussian setting, as compared to the toy dataset. The main reason is that in the toy dataset, there is a larger region where the gradients are in the $[1,0]^\top$ direction, which makes the decision boundary for signGD more linear, as well as a larger region where the neurons are only active for one of $\boldsymbol{\mu}_+$ or $\boldsymbol{\mu}_-$, which makes the decision boundary for GD more non-linear.

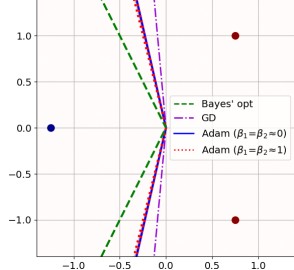

Figure 4: Comparison of the Bayes' optimal predictor and the predictors learned by two-layer NNs trained with GD, Adam ($\beta_1 = \beta_2 \approx 0$) or signGD, and Adam with $\beta_1 = \beta_2 \approx 1$ on the toy dataset (Gaussian dataset with $\sigma \to 0$).

# 4 Experimental Results

In this section, we present experimental results across synthetic and real-world datasets showing that GD exhibits simplicity bias while Adam promotes rich feature learning.

## 4.1 Gaussian Dataset

We consider the Gaussian data in Eq. (1) in this section, focusing on the *finite sample setting*, which is closer to practice as we train with the binary cross-entropy loss and consider Adam with momentum parameters $\beta_1 = \beta_2 = 0.9999$, although the results generalize to other values as well. We consider a small initialization scale and fix the outer layer weights. Specifically, $\boldsymbol{w}_k \sim \mathcal{N}(0, \frac{\alpha}{\sqrt{d}})$, and $a_k = \pm\frac{1}{\sqrt{m}}$ for $k \in [m]$, where $\alpha$ is a small constant.

Fig. 1 compares the decision boundaries learned by Adam and GD in the finite sample setting, with the Bayes' optimal predictor (for the population version of this setting). We set $n = 5000$, $m = 1000$, $\mu = 0.3, \omega = 2, \sigma_x = 0.2, \sigma_y = 0.15, \alpha = 0.001$, and use learning rates $0.1$ and $10^{-4}$ for GD and Adam, respectively. These results are similar to the population setting and show that the difference in the implicit bias of Adam and GD is quite robust to the choice of the training setting. For comparable train loss, the test accuracy of Adam is $0.32\%$ more than that of GD in this case. We also find that reducing $\mu$ increases the accuracy gap: repeating the same experiment with $\mu = 0.25$ leads to a gap of $0.595\%$. These results also generalize to settings where $d > 2$: with $m = 500$, $d = 20$ and $\mu = 0.25$, the gap is $0.203\%$. See Appendix B for additional results.

## 4.2 MNIST Dataset with Spurious Feature

In this section, we conduct an experiment to provide additional support for our theoretical results. We construct a binary classification task using $14 \times 14$ MNIST images, where we inject a $2 \times 2$ colored patch at the top left corner of the image (to model the simple feature). One class comprises digit '0' images with a red patch, while the other has digits '1' and '2' with a green patch. We train a two-layer NN using SGD or Adam. For test samples, we flip the patch color to check the model's reliance on the simple feature. We also train a linear model on this task, and measure agreement between its predictions on the test set with those of the NN trained with Adam/SGD. This serves as the measure of the complexity of the learned decision boundary. The results are shown in Table 2. We observe that the NN trained with Adam relies more on the digit features and generalizes better. It also has a complex/nonlinear decision boundary, as the agreement with the linear predictor is lower as compared to SGD. In addition, we plot the distribution of the train set margin for the two NNs as shown in Fig. 5. We observe that the margin for Adam is generally larger compared to SGD. These results support our theoretical findings that SGD learns simpler features compared to Adam.

|  | Test Accuracy | Agreement w/ Linear Model |
|---|---|---|
| SGD | 66.6 | 95.5 |
| Adam | 86 | 84 |

Table 2: Comparison of test accuracy and agreement with a linear model for a two-layer NN trained on MNIST with spurious correlation.

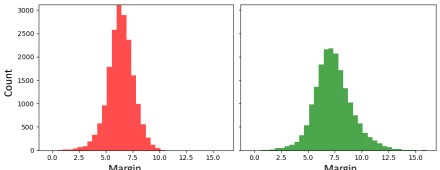

Figure 5: Distribution of margins of training-set samples from the MNIST dataset with spurious correlation for a two-layer NN trained using SGD (left) and Adam (right).

## 4.3 Dominoes Dataset

The Dominoes dataset [9, 12], specifically MNIST-CIFAR, is a binary image classification task. It contains images where the top half shows an MNIST digit [13] from classes $\{0, 1\}$, while the bottom half shows CIFAR-10 [14] images from classes {automobile, truck}. The MNIST portion corresponds to simple or spurious features, which are $95\%$ correlated with the label, while the CIFAR portion is the complex or core feature and is fully correlated. See Fig. 6 for example images from the dataset and Appendix B for further details.

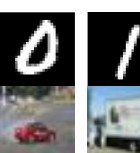

Figure 6: Example images from class $-1$ and $1$ from the MNIST-CIFAR dataset.

We define 4 groups based on the labels predicted by the core or the spurious feature. The minority groups correspond to images where the core and spurious features

disagree, and the majority groups correspond to images where they agree. The groups in the test set are balanced. The model can rely on the spurious feature to attain good train performance but can only generalize well on the balanced test set if it learns to use the core feature.

In Table 3, we report the average *original* worst-group accuracy on the balanced test set, for a ResNet-18 model (also see Appendix B, where we include results with ResNet-34 model and 99% spurious correlation). In addition, we report the average *core-only* worst-group accuracy on a test set where the spurious top half of the image is removed and replaced by a black image. This measures how much of the core features have been learned by the model. Lastly, we also report the average *decoded* worst-group accuracy, obtained by retraining the last layer of the model using logistic regression to fit a group balanced validation dataset, and then evaluating on the original test set. This gives a better evaluation for how much of the core features have been learned in the latent representation. We find that training with Adam leads to a significant gain across all three metrics as compared to training with SGD.

We also remark that these results (as well as those in Fig. 2) challenge the widely held consensus that SGD generally performs/generalizes better than Adam on image data [15]. We use ResNet-based models across the three image datasets and Adam leads to better worst-group accuracy across these cases. These results show that while SGD could be better for in-distribution generalization, Adam can be better for generalization under distribution shifts because it promotes richer feature learning.

| Optimizer | Original Acc. | Core-Only Acc. | Decoded Acc. |
|---|---|---|---|
| SGD | $0.81_{\pm0.38}$ | $1.66_{\pm1.79}$ | $71.04_{\pm0.63}$ |
| Adam | $\mathbf{14.17}_{\pm3.15}$ | $\mathbf{20.63}_{\pm5.75}$ | $\mathbf{84.66}_{\pm0.18}$ |

Table 3: Training with Adam leads to better worst-group accuracy on the original and core-only test sets, and after decoding, on the Dominoes (MNIST-CIFAR) dataset (95% spurious correlation), as compared to SGD, for a ResNet-18 model.

| Optimizer | Test Acc. | Decoded Core Corr. | Decoded Spurious Corr. |
|---|---|---|---|
| SGD | $89.58_{\pm1.92}$ | $0.51_{\pm0.08}$ | $0.78_{\pm0.08}$ |
| Adam | $\mathbf{97.87}_{\pm0.69}$ | $\mathbf{0.87}_{\pm0.03}$ | $\mathbf{0.36}_{\pm0.06}$ |

Table 4: Training with Adam leads to better test accuracy, decoded core, and decoded spurious correlations on the Boolean features dataset, as compared to SGD, for a three-layer NN.

## 4.4 Subgroup Robustness Datasets

We consider four benchmark subgroup robustness datasets, namely Waterbirds [16], CelebA [17], MultiNLI [18] and CivilComments [19, 20]. Each of these contains a core feature that is fully correlated with the label, and a spurious feature that is simpler but has a lower correlation. Waterbirds contains images of different birds on various backgrounds. The task is to classify whether the bird is a *landbird* or a *waterbird*. The background of the image is either *land* or *water*, and is spuriously correlated with the target label. CelebA consists of images of celebrity faces. The task is to classify whether the hair color is *blonde* or *not blonde*, and the gender of the celebrity *male* or *female* is the spurious feature. MultiNLI contains sentence pairs and the task is to predict how the second sentence relates to the first, out of three classes: *entailment*, *neutral*, and *contradiction*. The spurious features are *negation words* which are often but not always associated with contradiction. The CivilComments dataset consists of online comments and the task is to predict whether the comment is *toxic* or *non-toxic*. Toxicity is spuriously correlated with the mention of various *demographic attributes* in the comments, based on gender, race, or religion.

Prior work has shown that simplicity bias can be detrimental to worst-group test performance in the presence of spurious features [21]. The standard practice is to use SGD for image datasets and Adam(W) for language datasets. However, since Adam promotes richer feature learning, it should be more robust to spurious correlations across all datasets. Hence, we compare the performance of SGD and Adam on these datasets, when fine-tuning a pretrained BERT `bert-base-uncased` model [22] on the language datasets, and an ImageNet-pretrained ResNet-50 [23] model on the image datasets. To ensure fair comparison, we sweep optimizer-sepcific hyperparameters, namely learning rate, momentum, and weight decay (see Appendix B for details). Fig. 2 and Table 13 in Appendix B show the worst-group and average (group-balanced) accuracies on these datasets when training with SGD or Adam, based on the best worst-group validation accuracy. We see that training with Adam leads to significantly better worst-group accuracy as well as (slightly) better average accuracy compared to training with SGD.

### 4.5 Boolean Features Dataset

In this section, we consider the synthetic Boolean features dataset proposed by Qiu et al. [11] to study feature learning under spurious correlation. The dataset is designed to model the presence of two types of features: a set of complex core features with dimension $d_c$ that are fully correlated with the label, and a set of simple spurious features with dimension $d_s$ that have correlation strength $\lambda \in [0, 1]$ with the label. The rest of the $d_u = d - d_c - d_s$ features are uncorrelated with the label. For our experiments, the core and spurious features are modeled as staircase functions with degrees $d_c$ and $d_s < d_c$, respectively. Degree $d$ threshold staircase functions for a Boolean input $\boldsymbol{x} \in \{-1, +1\}^d$ are defined as:

$$f_{\text{staircase}}(\boldsymbol{x}) := \begin{cases} 1 & \text{if } x_1 + x_1 x_2 + \cdots + x_1 x_2 \ldots x_d \geq 0, \\ -1 & \text{otherwise.} \end{cases}$$

We train a three-layer NN using SGD and Adam. We consider $d = 50, d_c = 8, d_s = 1, \lambda = 0.9$. See Appendix B for a formal description of the dataset and further details of the experimental setting. To measure feature learning, we used the decoded core and spurious correlations [11, 24], which measure the extent to which the model has effectively learned the core and spurious features. The decoded core correlation is measured by retraining the last layer of the model using logistic regression to fit the core function and evaluating its correlations with $f_c$ on the uniform distribution, $\mathbb{E}_{\boldsymbol{x} \sim \text{Unif}(\{-1, 1\}^d)}[f_c(\boldsymbol{x})\text{sign}(f(\boldsymbol{x}))]$, where $f$ is the model. The decoded spurious correlation is measured similarly by retraining on the spurious function and measuring the correlation with $f_s$ on the test set.

We report the results in Table 4 evaluated at the lowest comparable training loss achieved by both optimizers. See Appendix B for the training curves and additional results. Adam records significantly higher average test accuracy and decoded core correlation, as well as lower decoded spurious correlation. This suggests that Adam's superior performance on the test set can be attributed to richer feature learning as it encourages the utilization of the core features and forgetting or down-weighting the spurious features. In contrast, SGD relies heavily on the simple spurious feature.

## 5 Related Work

In this section, we discuss related work on the implicit bias of GD, simplicity bias of NNs trained with GD, implicit bias of Adam and adaptive algorithms, and comparison of Adam and (S)GD in various settings.

**Implicit Bias of GD.** Since the pioneering studies that identified the implicit bias of linear classifiers on separable datasets [25], extensive research has been conducted on the implicit bias of gradient-based methods for linear models, NNs, and even self-attention models. Wang et al. [26] shows that GD with momentum exhibits the same implicit bias for linear models trained on separable data as vanilla GD. Nacson et al. [27], Ji and Telgarsky [28], Ji et al. [29] demonstrate fast convergence (in direction) of GD-based approaches with adaptive step-sizes to the $\ell_2$ max-margin predictor. It has also been shown that multilayer perceptrons (MLPs) trained with exponentially tailed loss functions on classification tasks, GD or gradient flow converge in direction to the KKT points of the max-margin problem in both finite [30, 31] and infinite-width [32] networks. Additionally, Phuong and Lampert [33], Frei et al. [34], Kou et al. [35] analyze the implicit bias of ReLU and Leaky-ReLU networks trained with GD on orthogonal data, while Mulayoff et al. [36] investigate convergence to stable minima. Other studies focus on the implicit bias to minimize rank in regression tasks using squared loss [37, 38, 39]. The recent survey Vardi [40] includes a comprehensive review of related work. More recently, Tarzanagh et al. [41, 42], Vasudeva et al. [43] studied single-head prompt and self-attention models with fixed linear decoder and characterized the implicit bias of attention weights trained with GD to converge to the solution of a hard-margin SVM problem.

**Simplicity Bias of NNs Trained with GD.** Kalimeris et al. [44] empirically demonstrate that NNs trained with SGD first learn to make predictions that are highly correlated with those of the best possible linear predictor for the task, and only later start to use more complex features to achieve further performance improvement. Shah et al. [9] created synthetic datasets and show that in the presence of 'simple' and 'complex' features (linearly separable vs non-linearly separable), (two-layer) NNs trained with SGD rely heavily on 'simple' features even when they have equal or even slightly worse predictive power than the 'complex' features. They also show that using SGD leads to learning small-margin and feature-impoverished classifiers, instead of large-margin and feature-dense classifiers, even on convergence, which contrasts with Kalimeris et al. [44].

**Implicit Bias of Adam and Other Adaptive Algorithms.** Wang et al. [45] show that homogeneous NNs trained with RMSprop or signGD converge to a KKT point of the $\ell_2$ max-margin problem, similar to GD, while AdaGrad has a different implicit bias. Zhang et al. [7] show that linear models trained on separable data with Adam converge to the $\ell_\infty$ max-margin solution. Recently Fan et al. [46] characterized the implicit bias of steepest descent algorithms for multiclass linearly separable data. Xie and Li [47] analyze loss minimization with AdamW and show that under some conditions, it converges to a KKT point of the $\ell_\infty$-norm constrained loss minimization.

**Adam vs (S)GD.** Zhou et al. [48] show that SGD converges to flatter minima while Adam converges to sharper minima. Andriushchenko et al. [49] show that flatter minima can correlate with better in-distribution generalization but may not be predictive of or even be negatively correlated with generalization under distribution shifts. Zou et al. [50] study an image-inspired dataset and show that CNNs trained with GD can generalize better than Adam. Ma et al. [51] show that adding noise to lower or higher frequency components of the data can lead to lower or higher robustness of Adam compared to GD. Kunstner et al. [52] show that the reason why Adam outperforms SGD on language data is because the performance of SGD deteriorates under heavy-tailed class imbalance, *i.e.*, when minority classes constitute a significant part of the data, whereas Adam is less sensitive and performs better. In contrast to their focus on multiple classes and training performance, our work focuses on generalization in a binary classification setting. Several works [53, 54, 55] also study why Adam outperforms SGD on attention models or Transformers.

# 6 Conclusion

In this work, we investigate the implicit bias of Adam and contrast it with (S)GD. NNs trained with SGD exhibit simplicity bias, whereas we find that training with Adam leads to richer feature learning, making the model more robust to spurious features and certain distribution shifts. We note that richer feature learning may not always be desirable; for instance, simplicity bias can be beneficial for better in-distribution generalization. However, it's important to characterize and contrast the implicit bias of Adam vs (S)GD in this context. To get a principled understanding, we identify a synthetic data setting with Gaussian clusters and theoretically show that two-layer ReLU NNs trained with GD or Adam on this task learn different solutions. GD exhibits simplicity bias and learns a linear predictor with a suboptimal margin, while Adam leads to richer feature learning and learns a nonlinear predictor that is closer to the Bayes' optimal predictor. Through theoretical and empirical results, our work adds to the conceptual understanding of how Adam works and poses important directions for future work, such as studying the implicit bias of Adam for other architectures, and the effect of weight decay on simple vs rich feature learning to study the implicit bias of AdamW.

## Acknowledgments

BV thanks Puneesh Deora for extensive discussions and feedback on the manuscript, and Surbhi Goel for helpful discussions. The authors acknowledge use of USC CARC's Discovery cluster, and thank the anonymous reviewers for their helpful comments. VS was supported by NSF CAREER Award CCF-2239265 and an Amazon Research Award. MS was supported by the Packard Fellowship in Science and Engineering, a Sloan Research Fellowship in Mathematics, an NSF CAREER Award #1846369, NSF-CIF awards #1813877 and #2008443, NSF SLES award #2417075, and NIH DP2LM014564-01. This work was done in part while BV, VS and MS were visiting the Simons Institute for the Theory of Computing at UC Berkeley.

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

# Appendix

## A  Omitted Proofs

The proof for Proposition 1 is as follows.

*Proof.* The optimal predictor can be found by solving for the following:

$$\tfrac{1}{2}\exp\left(-\tfrac{(x_1-\mu_1)^2}{2\sigma_x^2}-\tfrac{(x_2-\mu_2)^2}{2\sigma_y^2}\right)+\tfrac{1}{2}\exp\left(-\tfrac{(x_1-\mu_1)^2}{2\sigma_x^2}-\tfrac{(x_2+\mu_2)^2}{2\sigma_y^2}\right)=\exp\left(-\tfrac{(x_1+\mu_3)^2}{2\sigma_x^2}-\tfrac{x_2^2}{2\sigma_y^2}\right).$$

Simplification yields

$$0.5\left(1+\exp\left(-\tfrac{2\mu_2 x_2}{\sigma_y^2}\right)\right)=\exp\left(-\tfrac{(\mu_1+\mu_3)x_1}{\sigma_x^2}-\tfrac{\mu_2 x_2}{\sigma_y^2}+\tfrac{\mu_1^2-\mu_3^2}{2\sigma_x^2}+\tfrac{\mu_2^2}{2\sigma_y^2}\right).$$

Taking $\log$ on both sides and rearranging, we get:

$$\tfrac{(\mu_1+\mu_3)x_1}{\sigma_x^2}+\tfrac{\mu_2 x_2}{\sigma_y^2}=\tfrac{\mu_1^2-\mu_3^2}{2\sigma_x^2}+\tfrac{\mu_2^2}{2\sigma_y^2}-\log\left(0.5\left(1+\exp\left(-\tfrac{2\mu_2 x_2}{\sigma_y^2}\right)\right)\right).$$

For isotropic Gaussians, it simplifies to

$$(\mu_1+\mu_3)x_1+\mu_2 x_2=\tfrac{\mu_1^2+\mu_2^2-\mu_3^2}{2}-\sigma^2\log\left(0.5\left(1+\exp\left(-\tfrac{2\mu_2 x_2}{\sigma^2}\right)\right)\right).$$

Under realizability, we get

$$\omega x_1+x_2=-\tfrac{\sigma^2}{\mu}\log\left(0.5\left(1+\exp\left(-\tfrac{2\mu x_2}{\sigma^2}\right)\right)\right).$$

$\square$

### A.1  Gaussian Data

We can prove Proposition 2 as follows.

*Proof.* The population gradient can be simplified as follows.

$$
\begin{aligned}
\mathbb{E}[\mathbb{1}[\boldsymbol{w}^\top\boldsymbol{x}\geq 0]y\boldsymbol{x}] &= \mathbb{E}(\boldsymbol{x}|\boldsymbol{w}^\top\boldsymbol{x}\geq 0,y=1,\epsilon=1)\Pr[y=1]\Pr[\epsilon=1|y=1]\Pr[\boldsymbol{w}^\top\boldsymbol{x}\geq 0|y=1,\epsilon=1] \\
&\quad+\mathbb{E}(\boldsymbol{x}|\boldsymbol{w}^\top\boldsymbol{x}\geq 0,y=1,\epsilon=-1)\Pr[y=1]\Pr[\epsilon=-1|y=1]\Pr[\boldsymbol{w}^\top\boldsymbol{x}\geq 0|y=1,\epsilon=-1] \\
&\quad+\mathbb{E}(-\boldsymbol{x}|\boldsymbol{w}^\top\boldsymbol{x}\geq 0,y=-1)\Pr[y=-1]\Pr[\boldsymbol{w}^\top\boldsymbol{x}\geq 0|y=-1] \\
&= \frac{1}{4}\big(\Pr[\boldsymbol{w}^\top\boldsymbol{x}\geq 0|y=1,\epsilon=1]\mathbb{E}(\boldsymbol{x}|\boldsymbol{w}^\top\boldsymbol{x}\geq 0,y=1,\epsilon=1)+\Pr[\boldsymbol{w}^\top\boldsymbol{x}\geq 0|y=1,\epsilon=-1] \\
&\quad\mathbb{E}(\boldsymbol{x}|\boldsymbol{w}^\top\boldsymbol{x}\geq 0,y=1,\epsilon=-1)-2\Pr[\boldsymbol{w}^\top\boldsymbol{x}\geq 0|y=-1]\mathbb{E}(\boldsymbol{x}|\boldsymbol{w}^\top\boldsymbol{x}\geq 0,y=-1)\big).
\end{aligned}
$$

The conditional expectation $\mathbb{E}(\boldsymbol{x}'|\boldsymbol{w}^\top\boldsymbol{x}' \geq 0)$ can be simplified as follows. Let $\boldsymbol{\mu}' := \mathbb{E}(\boldsymbol{x}')$. Since we can write $\boldsymbol{x}' = \bar{\boldsymbol{w}}^\top\boldsymbol{x}'\bar{\boldsymbol{w}} + \bar{\boldsymbol{w}}_\perp^\top\boldsymbol{x}'\bar{\boldsymbol{w}}_\perp =: \boldsymbol{x}'_\| + \boldsymbol{x}'_\perp$, we have

$$\mathbb{E}(\boldsymbol{x}'|\boldsymbol{w}^\top\boldsymbol{x}' \geq 0) = \mathbb{E}(\boldsymbol{x}'_\||\boldsymbol{w}^\top\boldsymbol{x}' \geq 0) + \mathbb{E}(\boldsymbol{x}'_\perp|\boldsymbol{w}^\top\boldsymbol{x}' \geq 0)$$
$$= \mathbb{E}(\bar{\boldsymbol{w}}^\top\boldsymbol{x}'\bar{\boldsymbol{w}}|\boldsymbol{w}^\top\boldsymbol{x}' \geq 0) + \mathbb{E}(\boldsymbol{x}'_\perp)$$
$$= \frac{\boldsymbol{w}}{\|\boldsymbol{w}\|^2}\mathbb{E}(\boldsymbol{w}^\top\boldsymbol{x}'|\boldsymbol{w}^\top\boldsymbol{x}' \geq 0) + \mathbb{E}(\boldsymbol{x}') - \mathbb{E}(\boldsymbol{x}'_\|)$$
$$= \boldsymbol{\mu}' - \bar{\boldsymbol{w}}^\top\boldsymbol{\mu}'\bar{\boldsymbol{w}} + \frac{\boldsymbol{w}}{\|\boldsymbol{w}\|^2}\mathbb{E}(\boldsymbol{w}^\top\boldsymbol{x}'|\boldsymbol{w}^\top\boldsymbol{x}' \geq 0).$$

Using a result on the mean of truncated normal distribution from Burkardt [56], and that for a given $\boldsymbol{w}$, $\boldsymbol{w}^\top\boldsymbol{x}'$ is a Gaussian random variable, we have,

$$\mathbb{E}(\boldsymbol{w}^\top\boldsymbol{x}'|\boldsymbol{w}^\top\boldsymbol{x}' \geq 0) = \mu_{\boldsymbol{w}} + \sigma_{\boldsymbol{w}}\frac{\phi(-\frac{\mu_{\boldsymbol{w}}}{\sigma_{\boldsymbol{w}}})}{1 - \Phi(-\frac{\mu_{\boldsymbol{w}}}{\sigma_{\boldsymbol{w}}})},$$

where $\mu_{\boldsymbol{w}} := \boldsymbol{w}^\top\boldsymbol{\mu}'$, $\sigma_{\boldsymbol{w}} := \sigma\|\boldsymbol{w}\|$. Then, we have

$$\mathbb{E}(\boldsymbol{x}'|\boldsymbol{w}^\top\boldsymbol{x}' \geq 0) = \boldsymbol{\mu}' + \sigma\frac{\phi(-\frac{\mu_{\boldsymbol{w}}}{\sigma_{\boldsymbol{w}}})}{1 - \Phi(-\frac{\mu_{\boldsymbol{w}}}{\sigma_{\boldsymbol{w}}})}\bar{\boldsymbol{w}}.$$

Using the above, we can write the population gradient $-a\mathbb{E}[\mathbb{1}[\boldsymbol{w}^\top\boldsymbol{x} \geq 0]y\boldsymbol{x}]$ as:

$$-0.25a(p_+(\boldsymbol{\mu}_+ + \sigma\Gamma(\boldsymbol{\mu}_+, \bar{\boldsymbol{w}}_j)\bar{\boldsymbol{w}}) + p_-(\boldsymbol{\mu}_- + \sigma\Gamma(\boldsymbol{\mu}_-, \bar{\boldsymbol{w}})\bar{\boldsymbol{w}}) - 2p_0(\boldsymbol{\mu}_0 + \sigma\Gamma(\boldsymbol{\mu}_0, \bar{\boldsymbol{w}})\bar{\boldsymbol{w}})),$$

where $p_+ := \Pr[\boldsymbol{w}^\top\boldsymbol{x} \geq 0|y = 1, \epsilon = 1] = \Phi(\frac{\boldsymbol{\mu}_+^\top\bar{\boldsymbol{w}}}{\sigma})$, $p_- := \Pr[\boldsymbol{w}^\top\boldsymbol{x} \geq 0|y = 1, \epsilon = -1] = \Phi(\frac{\boldsymbol{\mu}_-^\top\bar{\boldsymbol{w}}}{\sigma})$, $p_0 := \Pr[\boldsymbol{w}^\top\boldsymbol{x} \geq 0|y = -1] = \Phi(\frac{\boldsymbol{\mu}_0^\top\bar{\boldsymbol{w}}}{\sigma})$, and $\Gamma(\boldsymbol{\mu}, \bar{\boldsymbol{w}}) := \frac{\phi(-\frac{\boldsymbol{\mu}^\top\bar{\boldsymbol{w}}}{\sigma})}{1-\Phi(-\frac{\boldsymbol{\mu}^\top\bar{\boldsymbol{w}}}{\sigma})} = \frac{\phi(\frac{\boldsymbol{\mu}^\top\bar{\boldsymbol{w}}}{\sigma})}{\Phi(\frac{\boldsymbol{\mu}^\top\bar{\boldsymbol{w}}}{\sigma})}$ (using the facts that for any $z$, $\phi(-z) = \phi(z)$ and $1 - \Phi(-z) = \Phi(z)$). Simplifying the expression then finishes the proof. $\qquad\square$

Next, we state the full version of Theorem 1 and prove it as follows.

**Theorem 5.** *(Full version of Theorem 1.) Consider the data in Eq. (1), neurons initialized such that $a_k = \pm 1$ with probability 0.5, $d = 2$, $\sigma_x = \sigma_y = \sigma$, $\omega \geq 2$ and $\lambda_0 := \frac{\mu}{\sigma} \geq 0.8$. Let $\boldsymbol{w}_{k,\infty} := \lim_{t\to\infty}\frac{\boldsymbol{w}_{k,t}}{t}$ and $\bar{\boldsymbol{w}}_{k,\infty} := \frac{\boldsymbol{w}_{k,\infty}}{\|\boldsymbol{w}_{k,\infty}\|}$, for $k \in [m]$. Then, the solution learned by gradient flow in the infinite sample setting with correlation loss is:*

$$\bar{\boldsymbol{w}}_{k,\infty} = a_k[1, 0]^\top.$$

*Proof.* For neuron $j \in [m]$, let $\theta_{j,t}$ denote the angle between $\boldsymbol{w}_{j,t}$ and the $x$-axis at iteration $t \geq 0$. We drop the subscript $j$ for convenience. Let $\bar{\boldsymbol{w}}_{GD}^* := [1, 0]^\top$. Then, $\cos\theta_t = \bar{\boldsymbol{w}}_t^\top\bar{\boldsymbol{w}}_{GD}^*$. We want to see if $\theta_t$ tends to 0 with time. Specifically, given $\theta \in [-\pi, \pi]$, we want to show that $a\frac{d\cos\theta_t}{dt} > 0$. We have:

$$a\frac{d\cos\theta_t}{dt} = a\dot{\bar{\boldsymbol{w}}}_t^\top\bar{\boldsymbol{w}}_{GD}^* = a\frac{\dot{\boldsymbol{w}}_t^\top}{\|\boldsymbol{w}_t\|}(I - \bar{\boldsymbol{w}}_t\bar{\boldsymbol{w}}_t^\top)\bar{\boldsymbol{w}}_{GD}^*$$

$$= \frac{a^2\sigma}{4\|\boldsymbol{w}_t\|}\left(\frac{\lambda(\omega^2-1)}{\omega^2+1}\left(\Phi(\lambda\bar{\boldsymbol{\mu}}_+^\top\bar{\boldsymbol{w}}_t) + \Phi(\lambda\bar{\boldsymbol{\mu}}_-^\top\bar{\boldsymbol{w}}_t)\right) + 2\lambda\Phi(\lambda\bar{\boldsymbol{\mu}}_0^\top\bar{\boldsymbol{w}}_t) + \frac{w_{t,1}}{\|\boldsymbol{w}_t\|}\left(\phi(\lambda\bar{\boldsymbol{\mu}}_+^\top\bar{\boldsymbol{w}}_t) + \phi(\lambda\bar{\boldsymbol{\mu}}_-^\top\bar{\boldsymbol{w}}_t) - 2\phi(\lambda\bar{\boldsymbol{\mu}}_0^\top\bar{\boldsymbol{w}}_t)\right)\right.$$

$$\left. - \frac{w_{t,1}}{\|\boldsymbol{w}_t\|}\left(\lambda(\Phi(\lambda\bar{\boldsymbol{\mu}}_+^\top\bar{\boldsymbol{w}}_t)\bar{\boldsymbol{\mu}}_+^\top\bar{\boldsymbol{w}}_t + \Phi(\lambda\bar{\boldsymbol{\mu}}_-^\top\bar{\boldsymbol{w}}_t)\bar{\boldsymbol{\mu}}_-^\top\bar{\boldsymbol{w}}_t - 2\Phi(\lambda\bar{\boldsymbol{\mu}}_0^\top\bar{\boldsymbol{w}}_t)\bar{\boldsymbol{\mu}}_0^\top\bar{\boldsymbol{w}}_t) + \left(\phi(\lambda\bar{\boldsymbol{\mu}}_+^\top\bar{\boldsymbol{w}}_t) + \phi(\lambda\bar{\boldsymbol{\mu}}_-^\top\bar{\boldsymbol{w}}_t) - 2\phi(\lambda\bar{\boldsymbol{\mu}}_0^\top\bar{\boldsymbol{w}}_t))\right)\right)$$

$$= \frac{a^2\sigma\lambda}{4\|\boldsymbol{w}_t\|}\left(\frac{(\omega^2-1)}{\omega^2+1}\frac{w_{t,2}^2}{\|\boldsymbol{w}_t\|^2}\left(\Phi(\lambda\bar{\boldsymbol{\mu}}_+^\top\bar{\boldsymbol{w}}_t) + \Phi(\lambda\bar{\boldsymbol{\mu}}_-^\top\bar{\boldsymbol{w}}_t)\right) - \frac{2\omega}{\omega^2+1}\frac{w_{t,1}w_{t,2}}{\|\boldsymbol{w}_t\|^2}\left(\Phi(\lambda\bar{\boldsymbol{\mu}}_+^\top\bar{\boldsymbol{w}}_t) - \Phi(\lambda\bar{\boldsymbol{\mu}}_-^\top\bar{\boldsymbol{w}}_t)\right) + 2\frac{w_{t,2}^2}{\|\boldsymbol{w}_t\|^2}\Phi(\lambda\bar{\boldsymbol{\mu}}_0^\top\bar{\boldsymbol{w}}_t)\right)$$

$$= \frac{a^2\sigma\lambda\sin^2\theta_t}{4\|\boldsymbol{w}_t\|}\left(\frac{(\omega^2-1)}{\omega^2+1}\left(\Phi(\lambda\bar{\boldsymbol{\mu}}_+^\top\bar{\boldsymbol{w}}_t) + \Phi(\lambda\bar{\boldsymbol{\mu}}_-^\top\bar{\boldsymbol{w}}_t)\right) - \frac{2\omega}{\omega^2+1}\frac{\cos\theta_t}{\sin\theta_t}\left(\Phi(\lambda\bar{\boldsymbol{\mu}}_+^\top\bar{\boldsymbol{w}}_t) - \Phi(\lambda\bar{\boldsymbol{\mu}}_-^\top\bar{\boldsymbol{w}}_t)\right) + 2\Phi(\lambda\bar{\boldsymbol{\mu}}_0^\top\bar{\boldsymbol{w}}_t)\right).$$

The first and third terms are always positive, so the sign depends on the second term. We note that the derivative is 0 when $\boldsymbol{w}_{t,2} = 0$, *i.e.*, $\theta_t = 0$. This indicates that once $\theta_t$ becomes 0, it remains 0.

Also, using the mean value theorem, we can write:

$$\Phi\left(\frac{\lambda((\omega^2-1)w_{t,1}+2\omega w_{t,2})}{\|\boldsymbol{w}_t\|(\omega^2+1)}\right) - \Phi\left(\frac{\lambda((\omega^2-1)w_{t,1}-2\omega w_{t,2})}{\|\boldsymbol{w}_t\|(\omega^2+1)}\right) = \phi(c)\frac{4\lambda\omega\sin\theta_t}{(\omega^2+1)},$$

for some $c \in \left[\frac{\lambda((\omega^2-1)w_{t,1}-2\omega w_{t,2})}{\|\boldsymbol{w}_t\|(\omega^2+1)}, \frac{\lambda((\omega^2-1)w_{t,1}+2\omega w_{t,2})}{\|\boldsymbol{w}_t\|(\omega^2+1)}\right]$. Clearly, $\phi(c) \le \phi(\lambda)$. The second term is lower bounded by $-\frac{8\lambda\omega^2}{(\omega^2+1)^2}\phi(\lambda)\cos\theta_t$. We now consider two cases:

Case 1: $\theta_t \in [-\pi, -\pi/2]$ or $\theta_t \in [\pi/2, \pi]$: In this case, $\cos\theta_t < 0$, so the second term, and hence the derivative, is positive.

Case 2: $\theta_t \in [-\pi/2, \pi/2]$: In this case, $\cos\theta_t > 0$, so the second term is negative, and we have to compare its magnitude to the other terms. Using $\Phi(\lambda\bar{\boldsymbol{\mu}}_+^\top\bar{\boldsymbol{w}}_t) + \Phi(\lambda\bar{\boldsymbol{\mu}}_-^\top\bar{\boldsymbol{w}}_t) \ge 1$ and $\bar{\boldsymbol{\mu}}_0^\top\bar{\boldsymbol{w}}_t \ge -1$, we have:

$$a\frac{d\cos\theta_t}{dt} \ge \frac{a^2\sigma\lambda\sin^2\theta_t}{4\|\boldsymbol{w}_t\|}\left(\frac{(\omega^2-1)}{\omega^2+1} - \frac{8\omega^2\lambda}{(\omega^2+1)^2}\cos\theta_t\phi(\lambda) + 2\Phi(-\lambda)\right).$$

Since $\cos\theta_t \le 1$ and $\Phi(-\lambda) > 0$, the RHS is positive when $\frac{\omega^2-1}{4\omega} \ge \frac{\mu}{\sigma}\phi(\lambda)$.

Let $E(\lambda_0, \omega) := \frac{\omega^2-1}{4\omega} - \lambda_0\phi(\lambda_0\frac{\omega^2+1}{2\omega})$.

$$\frac{dE}{d\lambda_0} = -\phi(\lambda_0\frac{\omega^2+1}{2\omega}) + \left(\lambda_0\frac{\omega^2+1}{2\omega}\right)^2\phi(\lambda_0\frac{\omega^2+1}{2\omega}) \ge 0,$$

when $\lambda_0 \ge \frac{2\omega}{\omega^2+1}$. The RHS here is a decreasing function of $\omega$ for $\omega \ge 2$. The condition becomes $\lambda_0 \ge 0.8$.

$$\frac{dE}{d\omega} = \frac{1}{4} + \frac{1}{4\omega^2} + \lambda_0^3\frac{\omega^2+1}{2\omega}\left(\frac{1}{2} - \frac{1}{2\omega^2}\right)\phi(\lambda_0\frac{\omega^2+1}{2\omega}) = \frac{\omega^2+1}{4\omega^2}\left(1 + \lambda_0^3\frac{\omega^4-1}{\omega}\phi(\lambda_0\frac{\omega^2+1}{2\omega})\right) \ge 0.$$

Since $E$ is an increasing function of both $\omega$ and $\lambda$, and we can numerically verify that $E(0.8, 2) > 0$, the result is true for all $\omega \ge 2$ and $\lambda_0 \ge 0.8$.

This shows that for neuron $\boldsymbol{w}_k$, $a_k\frac{d\cos\theta_{k,t}}{dt} \ge C\frac{(\sin\theta_{k,t})^2}{\|\boldsymbol{w}_{k,t}\|}$ for some constant $C > 0$.

Next, using Proposition 2, we can show that the gradients are bounded and consequently, the iterate norm is upper bounded as $\|\boldsymbol{w}_{k,t}\| \le c(t+1)$, for some constant $c > 0$. This gives $a_k\frac{d\cos\theta_{k,t}}{dt} \ge C'\frac{(\sin\theta_{k,t})^2}{t+1}$ for some constant $C' > 0$.

Next, consider $a_k = 1$, and suppose $\cos\theta_{k,t}$ stayed below some $L < 1$ for all $t$. Then, $(\sin\theta_{k,t})^2 \ge 1 - L^2 > 0$, so $\frac{d\cos\theta_{k,t}}{dt} \ge \frac{C'(1-L^2)}{(1+t)}$. Integrating both sides, we get $\cos\theta_{k,t} - \cos\theta_{k,0} \ge C'(1 - L^2)\log(1 + t)$, which diverges as $t \to \infty$, leading to a contradiction as $|\cos(\cdot)| \le 1$. The case where $a_k = -1$ follows similarly.

Hence, as $t \to \infty$, $\sin\theta_{k,t} \to 0$, and thus $\cos\theta_{k,t} \to \texttt{sign}(a_k)$. $\qquad\square$

Next, we state the full version of Theorem 2 and prove it as follows.

**Theorem 6.** *(Full version of Theorem 2.) Consider the data in Eq. (1), neurons initialized such that $a_k = \pm 1$ with probability 0.5, and $\sup_k\|\boldsymbol{w}_{k,0}\| < \eta/2$, $d = 2$, $\sigma_x = \sigma_y = \sigma$, $\omega \ge 2$ and $0.8 \le \frac{\mu}{\sigma} \le 1.5$. Let $\boldsymbol{w}_{k,\infty} := \lim_{t\to\infty}\frac{\boldsymbol{w}_{k,t}}{t}$ and $\bar{\boldsymbol{w}}_{k,\infty} := \frac{\boldsymbol{w}_{k,\infty}}{\|\boldsymbol{w}_{k,\infty}\|}$, for $k \in [m]$. Let $\theta_0$ denote the direction of $\boldsymbol{w}_{k,0}$. Then, the solution learned by signGD in the infinite sample setting with correlation loss is:*

$$\bar{\boldsymbol{w}}_{k,\infty} = \begin{cases} \frac{1}{\sqrt{2}}[1, 1]^\top & a_k > 0, \sin\theta_{k,0} > 0, \\ \frac{1}{\sqrt{2}}[1, -1]^\top & a_k > 0, \sin\theta_{k,0} < 0, \\ [1, 0]^\top & a_k > 0, \sin\theta_{k,0} = 0, \\ [-1, 0]^\top & a_k < 0. \end{cases}$$

*Proof.* For signGD, we can analyze the gradient expression for any $\boldsymbol{w}$:

$$\nabla_{\boldsymbol{w}}\widehat{L}(\boldsymbol{W}) = -\frac{a\sigma}{4}\Big(\Phi(\lambda\bar{\boldsymbol{\mu}}_+^\top\bar{\boldsymbol{w}})\lambda\bar{\boldsymbol{\mu}}_+ + \Phi(\lambda\bar{\boldsymbol{\mu}}_-^\top\bar{\boldsymbol{w}})\lambda\bar{\boldsymbol{\mu}}_- - 2\Phi(\lambda\bar{\boldsymbol{\mu}}_0^\top\bar{\boldsymbol{w}})\lambda\bar{\boldsymbol{\mu}}_0 + \big(\phi(\lambda\bar{\boldsymbol{\mu}}_+^\top\bar{\boldsymbol{w}}) + \phi(\lambda\bar{\boldsymbol{\mu}}_-^\top\bar{\boldsymbol{w}}) - 2\phi(\lambda\bar{\boldsymbol{\mu}}_0^\top\bar{\boldsymbol{w}})\big)\bar{\boldsymbol{w}}\Big).$$

Specifically, the gradient can be in the direction $[\pm 1, 0]^\top$ only when $[0,1]\nabla_{\boldsymbol{w}}\widehat{L}(\boldsymbol{W}) = 0$. We have:

$$[0,1]\nabla_{\boldsymbol{w}}\widehat{L}(\boldsymbol{W}) = -\tfrac{a\sigma}{4}\Big(\tfrac{2\omega\lambda}{\omega^2+1}\big(\Phi(\lambda\bar{\boldsymbol{\mu}}_+^\top\boldsymbol{w}) - \Phi(\lambda\bar{\boldsymbol{\mu}}_-^\top\boldsymbol{w})\big) + \sin\theta(\phi(\lambda\bar{\boldsymbol{\mu}}_+^\top\boldsymbol{w}) + \phi(\lambda\bar{\boldsymbol{\mu}}_-^\top\boldsymbol{w}) - 2\phi(\lambda\bar{\boldsymbol{\mu}}_0^\top\boldsymbol{w}))\Big)$$

$$= -\tfrac{a\sigma\sin\theta}{4}\Big(2\big(\tfrac{2\omega\lambda}{\omega^2+1}\big)^2\phi(c) + \phi(\lambda\bar{\boldsymbol{\mu}}_+^\top\boldsymbol{w}) + \phi(\lambda\bar{\boldsymbol{\mu}}_-^\top\boldsymbol{w}) - 2\phi(\lambda\bar{\boldsymbol{\mu}}_0^\top\boldsymbol{w})\Big),$$

where $c \in [\lambda\bar{\boldsymbol{\mu}}_-^\top\bar{\boldsymbol{w}}, \lambda\bar{\boldsymbol{\mu}}_+^\top\bar{\boldsymbol{w}}]$. Consider the expression in the parenthesis. Assuming $\tfrac{\omega^2-1}{2\omega} \geq 1$, we have:

$$2\big(\tfrac{2\omega\lambda}{\omega^2+1}\big)^2\phi(c) + \phi(\lambda\bar{\boldsymbol{\mu}}_+^\top\boldsymbol{w}) + \phi(\lambda\bar{\boldsymbol{\mu}}_-^\top\boldsymbol{w}) - 2\phi(\lambda\bar{\boldsymbol{\mu}}_0^\top\boldsymbol{w}) \geq 2\Big(\big(\big(\tfrac{2\omega\lambda}{\omega^2+1}\big)^2 + 1\big)\phi(\tfrac{2\omega\lambda}{\omega^2+1}) - \phi(0)\Big)$$

$$= 2\Big(\big(\big(\tfrac{\mu}{\sigma}\big)^2 + 1\big)\phi(\tfrac{\mu}{\sigma}) - \phi(0)\Big) > 0,$$

whenever $\tfrac{\mu}{\sigma} \leq 1.5$ (we can check this numerically, and use the fact that $\phi(z)$ is a decreasing function of $z \geq 0$).

Thus, the gradient is only in the $[\pm 1, 0]^\top$ direction when $\sin\theta = 0$, *i.e.*, when $\boldsymbol{w}$ is in that direction.

Next, we can check if there are neurons in the $[0, \pm 1]^\top$ direction. We have:

$$[1,0]\nabla_{\boldsymbol{w}}\widehat{L}(\boldsymbol{W}) = -\tfrac{a\sigma}{4}\Big(2\lambda\tfrac{\omega^2-1}{\omega^2+1}\big(\Phi(\lambda\bar{\boldsymbol{\mu}}_+^\top\boldsymbol{w}) + \Phi(\lambda\bar{\boldsymbol{\mu}}_-^\top\boldsymbol{w})\big) + 2\lambda\Phi(\lambda\bar{\boldsymbol{\mu}}_0^\top\boldsymbol{w}) + \cos\theta(\phi(\lambda\bar{\boldsymbol{\mu}}_+^\top\boldsymbol{w}) + \phi(\lambda\bar{\boldsymbol{\mu}}_-^\top\boldsymbol{w}) - 2\phi(\lambda\bar{\boldsymbol{\mu}}_0^\top\boldsymbol{w}))\Big).$$

The expression in the parenthesis is positive as long as $\lambda\tfrac{\omega^2-1}{\omega^2+1} + 0.5\lambda \geq 0.4$, or $\tfrac{3\omega^2-1}{1.6\omega} \geq \tfrac{\sigma}{\mu}$. Let $E(\lambda_0, \omega) = \lambda_0 - \tfrac{1.6\omega}{3\omega^2-1}$. We can show that it is an increasing function of both $\lambda_0$ and $\omega$. Since $E(0.8, 2) > 0$, the result holds for all $\omega \geq 2$ and $\lambda_0 \geq 0.8$.

Based on these calculations, the updates are along $[\pm 1, \pm 1]^\top$ directions, depending on the sign of $a$ and $\sin\theta$. Specifically, we have four cases as shown in Table 5 for $\theta_t$ and $\theta_{t+1}$ at any $t$.

| $\texttt{sign}(\sin\theta_t)$ | $\texttt{sign}(a)$ | $\texttt{sign}(-[0,1]\nabla_{\boldsymbol{w}}\widehat{L}(\boldsymbol{W}))$ | $\texttt{sign}(\sin\theta_{t+1})$ |
|---|---|---|---|
| +ve | +ve | +ve | +ve |
| +ve | -ve | -ve | -ve |
| -ve | +ve | -ve | -ve |
| -ve | -ve | +ve | +ve |

Table 5: Different cases for $\theta_t$ and $\theta_{t+1}$ in the analysis of signGD.

Using the small initialization condition, the first iterate is dominated by the update direction: $\boldsymbol{w}_{k,1} = \boldsymbol{w}_{k,0} - \eta\boldsymbol{s}_0$, where $\boldsymbol{s}_0 = \texttt{sign}(\nabla\widehat{L}(\boldsymbol{w}_{k,0}))$, so for each coordinate $i$, $|w_{k,1,i}| \geq \eta - \eta/2 = \eta/2$, hence $\boldsymbol{w}_{k,1}$ is sign-aligned with $-\boldsymbol{s}_0$. Consequently, the next update follows from Table 5, and the argument proceeds by recursion.

This shows that whenever $a > 0$, the updates for neurons in the first/second or third/fourth quadrant are along $[1, 1]^\top$ or $[1, -1]^\top$, respectively. However, when $a < 0$, the updates for neurons in the first/second or third/fourth quadrants alternate between $[-1, \pm 1]^\top$ and $[-1, \mp 1]^\top$. As a result, at even iterations, these neurons are close to the $[-1, 0]^\top$ direction (but may not be exactly aligned due to the initialization). However, in the limit $t \to \infty$, these neurons converge in this direction. $\qquad\square$

We state the full version of Theorem 3 and prove it below.

**Theorem 7.** *(Full version of Theorem 3.) Consider the data in Eq. (1) with $d = 2$ and $\omega \in [2, 12]$, $\kappa = \tfrac{\sigma_x^2}{\sigma_y^2} \in [\tfrac{1}{\omega^2}, 1]$, $\tfrac{\mu}{\sigma_y} \geq 0.8\sqrt{\kappa}\omega$. Consider two predictors,*

$$\textit{Linear: } \hat{y} = \texttt{sign}(x_1),$$
$$\textit{Piece-wise Linear: } \hat{y}' = \begin{cases} \texttt{sign}(3x_1 + x_2) & x_2 \geq 0, \\ \texttt{sign}(3x_1 - x_2) & x_2 < 0. \end{cases}$$

*Then, it holds that $\mathbb{E}(\hat{y}' \neq y) - \mathbb{E}(\hat{y} \neq y) < 0$.*

*Proof.* Linear: $\hat{y} = \texttt{sign}(ax_1 + bx_2)$. Piece-wise Linear: $\hat{y}' = \begin{cases} \texttt{sign}(ax_1 + bx_2) & x_2 \geq 0, \\ \texttt{sign}(ax_1 - bx_2) & x_2 < 0. \end{cases}$

$$\mathbb{E}(\hat{y} \neq y) = \frac{1}{4}\left[\Phi\left(-\frac{\mu\left(\frac{a}{2}\left(\kappa\omega - \frac{1}{\omega}\right)+b\right)}{\sigma_y\sqrt{a^2\kappa+b^2}}\right) + \Phi\left(-\frac{\mu\left(\frac{a}{2}\left(\kappa\omega - \frac{1}{\omega}\right)-b\right)}{\sigma_y\sqrt{a^2\kappa+b^2}}\right)\right] + \frac{1}{2}\Phi\left(-\frac{\mu\frac{a}{2}\left(\kappa\omega+\frac{1}{\omega}\right)}{\sqrt{\kappa}\,\sigma_y}\right).$$

When $a = 1, b = 0$, we get:

$$\mathbb{E}(\hat{y} \neq y) = \frac{1}{2}\Phi\left(-\frac{\mu\left(\kappa\omega - \frac{1}{\omega}\right)}{2\sigma_y\sqrt{\kappa}}\right) + \frac{1}{2}\Phi\left(-\frac{\mu\left(\kappa\omega + \frac{1}{\omega}\right)}{2\sigma_y\sqrt{\kappa}}\right).$$

Considering the non-isotropic case, when $a = 3, b = 1$, we have:

$$\mathbb{E}(\hat{y}' \neq y) < \frac{1}{2}\Phi\left(-\frac{\mu}{\sqrt{9\kappa+1}\sigma_y}\frac{3\kappa\omega^2-3+2\omega}{2\omega}\right) + \Phi\left(-\frac{3\mu}{\sqrt{\kappa}\sigma_y}\frac{\kappa\omega^2+1}{2\omega}\right)$$

$$\implies \mathbb{E}(\hat{y}' \neq y) - \mathbb{E}(\hat{y} \neq y) < \frac{1}{2}\Phi\left(-\frac{\mu}{\sqrt{9\kappa+1}\sigma_y}\frac{3\kappa\omega^2-3+2\omega}{2\omega}\right) + \frac{1}{2}\Phi\left(-\frac{3\mu}{\sqrt{\kappa}\sigma_y}\frac{\kappa\omega^2+1}{2\omega}\right) - \frac{1}{2}\Phi\left(-\frac{\mu}{\sqrt{\kappa}\sigma_y}\frac{\kappa\omega^2-1}{2\omega}\right)$$

$$= \frac{1}{2}\left(\Phi\left(-\frac{\mu}{\sqrt{9\kappa+1}\sigma_y}\frac{3\kappa\omega^2-3+2\omega}{2\omega}\right) + \Phi\left(\frac{\mu}{\sqrt{\kappa}\sigma_y}\frac{\kappa\omega^2-1}{2\omega}\right) - \Phi\left(\frac{3\mu}{\sqrt{\kappa}\sigma_y}\frac{\kappa\omega^2+1}{2\omega}\right)\right) =: 0.5E(\lambda_0, \kappa, \omega),$$

where $\lambda_0 := \frac{\mu}{\sigma_y}$. We can analyze the first derivatives:

$$\frac{dE}{d\omega} = \frac{\lambda_0}{2\omega^2}\left(-\frac{3(\kappa\omega^2+1)}{\sqrt{9\kappa+1}}\phi\left(\frac{\lambda_0}{\sqrt{9\kappa+1}}\frac{3\kappa\omega^2-3+2\omega}{2\omega}\right) + \frac{(\kappa\omega^2+1)}{\sqrt{\kappa}}\phi\left(\frac{\lambda_0}{\sqrt{\kappa}}\frac{\kappa\omega^2-1}{2\omega}\right) - \frac{3(\kappa\omega^2-1)}{\sqrt{\kappa}}\phi\left(\frac{3\lambda_0}{\sqrt{\kappa}}\frac{\kappa\omega^2+1}{2\omega}\right)\right)$$

$$= \frac{\lambda_0(\kappa\omega^2+1)}{2\omega^2\sqrt{\kappa}}\phi\left(\frac{\lambda_0}{\sqrt{\kappa}}\frac{\kappa\omega^2-1}{2\omega}\right)\left(1 - \frac{3\sqrt{\kappa}}{\sqrt{9\kappa+1}}\exp\left(-\frac{\lambda_0^2}{8\omega^2}\left(\frac{(3\kappa\omega^2-3+2\omega)^2}{9\kappa+1} - \frac{(\kappa\omega^2-1)^2}{\kappa}\right)\right) - \frac{3(\kappa\omega^2-1)}{(\kappa\omega^2+1)}\exp\left(-\frac{\lambda_0^2(9(\kappa\omega^2+1)^2-(\kappa\omega^2-1)^2)}{8\omega^2\kappa}\right)\right)$$

$$= \frac{\lambda_0(\kappa\omega^2+1)}{2\omega^2\sqrt{\kappa}}\phi\left(\frac{\lambda_0}{\sqrt{\kappa}}\frac{\kappa\omega^2-1}{2\omega}\right)\left(1 - \frac{3\sqrt{\kappa}}{\sqrt{9\kappa+1}}\exp\left(-\frac{\lambda_0^2(\kappa(3\kappa\omega^2-3+2\omega)^2-(9\kappa+1)(\kappa\omega^2-1)^2)}{8\kappa\omega^2(9\kappa+1)}\right) - \frac{3(\kappa\omega^2-1)}{(\kappa\omega^2+1)}\exp\left(-\frac{\lambda_0^2(2\kappa^2\omega^4+2+5\kappa\omega^2)}{2\omega^2\kappa}\right)\right).$$

This is positive when the following two conditions hold:

$$\left(\kappa(3\kappa\omega^2 - 3 + 2\omega)^2 - (9\kappa+1)(\kappa\omega^2-1)^2\right) = \kappa(9\kappa^2\omega^4 + 9 + 4\omega^2 - 18\kappa\omega^2 - 12\omega + 12\kappa\omega^3) - (9\kappa+1)(\kappa^2\omega^4 + 1 - 2\kappa\omega^2)$$

$$= \kappa^2(12\omega^3 - \omega^4) + \kappa(6\omega^2 - 12\omega) - 1 > 0,$$

when $\kappa \geq \frac{-(3\omega-6)+\sqrt{(3\omega-6)^2+(12\omega-\omega^2)}}{(12\omega^2-\omega^3)} = \frac{-(3\omega-6)+\sqrt{8\omega^2-24\omega+36}}{(12\omega^2-\omega^3)} > 0$ since $\omega \leq 12$. $\kappa \leq 1$ implies $12\omega^3 - \omega^4 + 6\omega^2 - 12\omega - 1 \geq 0$. Since $\lambda_0 \geq 0.8\sqrt{\kappa}\omega$, and $\omega \in [2, 12]$, we have:

$$1 - \frac{3\sqrt{\kappa}}{\sqrt{9\kappa+1}}\exp\left(-\frac{\lambda_0^2(\kappa^2(12\omega^3-\omega^4)+\kappa(6\omega^2-12\omega)-1)}{8\kappa\omega^2(9\kappa+1)}\right) - \frac{3(\kappa\omega^2-1)}{(\kappa\omega^2+1)}\exp\left(-\frac{\lambda_0^2(2\kappa^2\omega^4+2+5\kappa\omega^2)}{2\omega^2\kappa}\right)$$

$$\geq 1 - \frac{3}{\sqrt{10}}\exp\left(-\frac{0.8^2 79}{80}\right) - 3\frac{143}{145}\exp\left(-\frac{0.8^2 54}{2}\right) > 0.$$

Next, we compute the derivative wrt $\lambda_0$.

$$\frac{dE}{d\lambda_0} = \frac{1}{2\omega}\left(-\frac{3\kappa\omega^2-3+2\omega}{\sqrt{9\kappa+1}}\phi\left(\frac{\lambda_0}{\sqrt{9\kappa+1}}\frac{3\kappa\omega^2-3+2\omega}{2\omega}\right) + \frac{(\kappa\omega^2-1)}{\sqrt{\kappa}}\phi\left(\frac{\lambda_0}{\sqrt{\kappa}}\frac{\kappa\omega^2-1}{2\omega}\right) - \frac{3(\kappa\omega^2+1)}{\sqrt{\kappa}}\phi\left(\frac{3\lambda_0}{\sqrt{\kappa}}\frac{\kappa\omega^2+1}{2\omega}\right)\right)$$

$$= \frac{(\kappa\omega^2-1)}{2\omega\sqrt{\kappa}}\phi\left(\frac{\lambda_0}{\sqrt{\kappa}}\frac{\kappa\omega^2-1}{2\omega}\right)\left(1 - \frac{\sqrt{\kappa}(3\kappa\omega^2-3+2\omega)}{\sqrt{9\kappa+1}(\kappa\omega^2-1)}\exp\left(-\frac{\lambda_0^2(\kappa^2(12\omega^3-\omega^4)+\kappa(6\omega^2-12\omega)-1)}{8\omega^2\kappa(9\kappa+1)}\right) - \frac{3(\kappa\omega^2+1)}{(\kappa\omega^2-1)}\exp\left(-\frac{\lambda_0^2(2\kappa^2\omega^4+2+5\kappa\omega^2)}{8\omega^2\kappa}\right)\right)$$

Since $\kappa\omega^2 \geq 1$,

$$1 - \frac{\sqrt{\kappa}(3\kappa\omega^2-3+2\omega)}{\sqrt{9\kappa+1}(\kappa\omega^2-1)}\exp\left(-\frac{\lambda_0^2(\kappa^2(12\omega^3-\omega^4)+\kappa(6\omega^2-12\omega)-1)}{8\omega^2\kappa(9\kappa+1)}\right) - \frac{3(\kappa\omega^2+1)}{(\kappa\omega^2-1)}\exp\left(-\frac{\lambda_0^2(2\kappa^2\omega^4+2+5\kappa\omega^2)}{2\omega^2\kappa}\right)$$

$$\geq 1 - \frac{3}{\sqrt{10}}\frac{\omega^2-1+2\omega/3}{\omega^2-1}\exp\left(-\frac{0.8^2 79}{80}\right) - 3\frac{\omega^2+1}{\omega^2-1}\exp\left(-\frac{0.8^2 54}{2}\right) \geq 1 - \frac{3}{\sqrt{10}}\frac{13}{9}\exp\left(-\frac{0.8^2 79}{80}\right) - 3\frac{5}{3}\exp\left(-\frac{0.8^2 54}{2}\right) > 0.$$

Next, we compute the derivative wrt $\kappa$.

$$\frac{dE}{d\kappa} = \frac{\lambda_0}{4\omega}\left(-\frac{51\kappa\omega^2+6\omega^2-2\omega+3}{(9\kappa+1)\sqrt{9\kappa+1}}\phi\left(\frac{\lambda_0}{\sqrt{9\kappa+1}}\frac{3\kappa\omega^2-3+2\omega}{2\omega}\right) + \frac{(\kappa\omega^2+1)}{\kappa\sqrt{\kappa}}\phi\left(\frac{\lambda_0}{\sqrt{\kappa}}\frac{\kappa\omega^2-1}{2\omega}\right) - \frac{3(\kappa\omega^2-1)}{\kappa\sqrt{\kappa}}\phi\left(\frac{3\lambda_0}{\sqrt{\kappa}}\frac{\kappa\omega^2+1}{2\omega}\right)\right)$$

$$= \frac{\lambda_0(\kappa\omega^2+1)}{4\omega\kappa\sqrt{\kappa}}\phi\left(\frac{\lambda_0}{\sqrt{\kappa}}\frac{\kappa\omega^2-1}{2\omega}\right)\left(1 - \frac{\kappa\sqrt{\kappa}(51\kappa\omega^2+6\omega^2-2\omega+3)}{(9\kappa+1)\sqrt{9\kappa+1}(\kappa\omega^2+1)}\exp\left(\frac{-\lambda_0^2(\kappa^2(12\omega^3-\omega^4)+\kappa(6\omega^2-12\omega)-1)}{8\omega^2\kappa(9\kappa+1)}\right)\right.$$

$$\left. - \frac{3(\kappa\omega^2-1)}{(\kappa\omega^2+1)}\exp\left(\frac{-\lambda_0^2(2\kappa^2\omega^4+2+5\kappa\omega^2)}{8\omega^2\kappa}\right)\right).$$

The expression in the parenthesis is lower bounded as:

$$1 - \frac{1}{10^{3/2}}\frac{57\omega^2-2\omega+3}{\omega^2+1}\exp\left(-\frac{0.8^2 79}{80}\right) - 3\frac{143}{145}\exp\left(-\frac{0.8^2 54}{2}\right) \geq 1 - \frac{1}{10^{3/2}}\frac{57(144)-21}{145}\exp\left(-\frac{0.8^2 79}{80}\right) - 3\frac{143}{145}\exp\left(-\frac{0.8^2 54}{2}\right) > 0.$$

$\square$

## A.2 Toy Data

We can write the Bayes' optimal predictor in the toy setting as follows.

We consider three datapoints $(\boldsymbol{x}, y)$: $([-\mu_3, 0]^\top, -1)$, $([\mu_1, \mu_2]^\top, 1)$ and $([\mu_1, -\mu_2]^\top, 1)$, where $\mu_1, \mu_2, \mu_3 > 0$. The optimal predictor can be found by solving the following:

$$\min(\sqrt{(x_1 - \mu_1)^2 + (x_2 - \mu_2)^2}, \sqrt{(x_1 - \mu_1)^2 + (x_2 + \mu_2)^2}) = \sqrt{(x_1 + \mu_3)^2 + x_2^2}.$$

Solving this gives a piecewise linear function:

$$(\mu_1 + \mu_3)x_1 + \mu_2 x_2 = \frac{\mu_1^2 + \mu_2^2 - \mu_3^2}{2}, \quad x_2 > 0$$
$$(\mu_1 + \mu_3)x_1 - \mu_2 x_2 = \frac{\mu_1^2 + \mu_2^2 - \mu_3^2}{2}, \quad x_2 \le 0.$$

In the realizable setting, this is:

$$\omega x_1 + x_2 = 0, \quad x_2 > 0$$
$$\omega x_1 - x_2 = 0, \quad x_2 \le 0.$$

We now state the full version of Theorem 4, followed by the proof.

**Theorem 8** (Full version of Theorem 4). *Consider* $\|\boldsymbol{w}_k\| \le \frac{\eta\mu}{8\omega(\omega^2-1)}(((3\omega^2 + 1)(\omega^2 - 1) - 4\omega^2) \wedge (4\omega^2 - (\omega^2 - 1)^2) \wedge \frac{8\omega}{\mu}(2\omega + 1 - \omega^2))$ *and* $1 + \frac{2}{\sqrt{3}} < \omega^2 < 3 + 2\sqrt{2}$. *Let* $\bar{\boldsymbol{w}}_{k,\infty} := \frac{\lim_{t\to\infty} \frac{\boldsymbol{w}_{k,t}}{t}}{\|\lim_{t\to\infty} \frac{\boldsymbol{w}_{k,t}}{t}\|}$, *for neuron* $k \in [m]$ *and* $p := \frac{\tan^{-1} \frac{\omega^2-1}{2\omega}}{\pi}$. *Then, for* $m \to \infty$, *the solutions learned by GD, signGD, and Adam are as shown in Table 6,*

| | GD | | Adam ($\beta_1 = \beta_2 = 0$) or signGD | | Adam ($\beta_1 = \beta_2 \approx 1$) | |
|---|---|---|---|---|---|---|
| $\bar{\boldsymbol{w}}_{k,\infty} =$ | $\begin{cases} [1,0]^\top \\ [-1,0]^\top \\ \frac{1}{\omega^2+1}[\omega^2 - 1, 2\omega]^\top \\ \frac{1}{\omega^2+1}[\omega^2 - 1, -2\omega]^\top \end{cases}$ | $\begin{array}{l} \text{w.p. } \frac{1}{4} + \frac{p}{2} \\ \text{w.p. } \frac{1}{2} \\ \text{w.p. } \frac{1}{8} - \frac{p}{4} \\ \text{w.p. } \frac{1}{8} - \frac{p}{4} \end{array}$ | $\begin{cases} [1,0]^\top \\ [-1,0]^\top \\ \frac{1}{\sqrt{2}}[1,1]^\top \\ \frac{1}{\sqrt{2}}[1,-1]^\top \end{cases}$ | $\begin{array}{l} \text{w.p. } p \\ \text{w.p. } \frac{1}{2} \\ \text{w.p. } \frac{1}{4} - \frac{p}{2} \\ \text{w.p. } \frac{1}{4} - \frac{p}{2} \end{array}$ | $\begin{cases} [1,0]^\top \\ [-1,0]^\top \\ \frac{1}{\sqrt{2}}[1,1]^\top \\ \frac{1}{\sqrt{2}}[1,-1]^\top \\ \frac{1}{\sqrt{s^2+1}}[s,1]^\top \\ \frac{1}{\sqrt{s^2+1}}[s,-1]^\top \end{cases}$ | $\begin{array}{l} \text{w.p. } p \\ \text{w.p. } \frac{1}{2} \\ \text{w.p. } \frac{1}{8} - \frac{p}{4} \\ \text{w.p. } \frac{1}{8} - \frac{p}{4} \\ \text{w.p. } \frac{1}{8} - \frac{p}{4} \\ \text{w.p. } \frac{1}{8} - \frac{p}{4} \end{array}$ |

Table 6: Solutions learned by GD, signGD and Adam (see Theorem 4).

*where* $s$ *is a constant between* $0.72$ *and* $1$. *In each case, the sign of the first element of* $\boldsymbol{w}_{k,\infty}$ *is the same as* $sign(a_k)$.

*Proof.* Let $z := (\boldsymbol{x}, y)$, and $\bar{z}_1 := -\frac{\mu}{2}[\omega + \frac{1}{\omega}, 0]$, $\bar{z}_2 := \frac{\mu}{2}[\omega - \frac{1}{\omega}, 2]$, $\bar{z}_3 := \frac{\mu}{2}[\omega - \frac{1}{\omega}, -2]$. Define three sets $S_1, S_2, S_3$ as $S_1 := \{z \in S : x_1 < 0\}$, $S_2 := \{z \in S : x_2 > 0\}$, $S_3 := \{z \in S : x_2 < 0\}$.

**First iteration.** We first analyze the gradients at the first iteration. Consider different cases where $\boldsymbol{w}_{k,0}^\top \boldsymbol{x} \ge 0$ depending on different samples $\boldsymbol{x}$. Table 7 lists the population gradients depending on which samples contribute to the gradient. See Fig. 7 for an illustration. Note that $\theta = \tan^{-1} \frac{\mu_2}{\mu_1} = \tan^{-1} \frac{2\omega}{\omega^2-1}$, and $\frac{\pi}{2} - \theta = \tan^{-1} \frac{\omega^2-1}{2\omega}$.

| Set $S$ s.t. $\boldsymbol{w}_{k,0}^\top \boldsymbol{x} > 0$ | Pop. Gradient $\mathbb{E}_{z\sim\mathcal{D}}[y\boldsymbol{x}|\boldsymbol{x}\in S]$ | Prob. of such $\boldsymbol{w}_k$ |
|---|---|---|
| $S_2 \cup S_3$ | $\frac{1}{2}[\mu_1, 0]^\top$ | $\frac{\tan^{-1}\frac{\omega^2-1}{2\omega}}{\pi}$ |
| $S_2$ | $\frac{1}{4}[\mu_1, \mu_2]^\top$ | $\frac{\frac{\pi}{2}-\tan^{-1}\frac{\omega^2-1}{2\omega}}{2\pi}$ |
| $S_1 \cup S_2$ | $\frac{1}{4}[\mu_1 + 2\mu_3, \mu_2]^\top$ | $\frac{\frac{\pi}{2}-\tan^{-1}\frac{\omega^2-1}{2\omega}}{2\pi}$ |
| $S_1$ | $\frac{1}{2}[\mu_3, 0]^\top$ | $\frac{\tan^{-1}\frac{\omega^2-1}{2\omega}}{\pi}$ |
| $S_3 \cup S_1$ | $\frac{1}{4}[\mu_1 + 2\mu_3, -\mu_2]^\top$ | $\frac{\frac{\pi}{2}-\tan^{-1}\frac{\omega^2-1}{2\omega}}{2\pi}$ |
| $S_3$ | $\frac{1}{4}[\mu_1, -\mu_2]^\top$ | $\frac{\frac{\pi}{2}-\tan^{-1}\frac{\omega^2-1}{2\omega}}{2\pi}$ |

Table 7: Population gradients and corresponding probabilities depending on the region of initialization of the neurons.

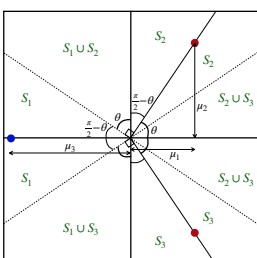

Figure 7: An illustration of the toy dataset and the set $S$ such that $\boldsymbol{w}_{k,0}^\top \boldsymbol{x} > 0$ depending on the region of initialization.

Using the population gradients in Table 7, the updates for the different algorithms, are written as:

$$\text{GD:}\quad \boldsymbol{w}_{k,1} = \boldsymbol{w}_{k,0} + \frac{a_k\eta\mu}{4}\begin{cases} \left[\frac{\omega^2-1}{\omega}, 0\right]^\top & \text{w.p. } \frac{\tan^{-1}\frac{\omega^2-1}{2\omega}}{\pi} \\[2mm] \left[\frac{\omega^2+1}{\omega}, 0\right]^\top & \text{w.p. } \frac{\tan^{-1}\frac{\omega^2-1}{2\omega}}{\pi} \\[2mm] \left[\frac{\omega^2-1}{2\omega}, 1\right] & \text{w.p. } \frac{1}{4}-\frac{\tan^{-1}\frac{\omega^2-1}{2\omega}}{2\pi} \\[2mm] \left[\frac{\omega^2-1}{2\omega}, -1\right] & \text{w.p. } \frac{1}{4}-\frac{\tan^{-1}\frac{\omega^2-1}{2\omega}}{2\pi} \\[2mm] \left[\frac{3\omega^2+1}{2\omega}, 1\right]^\top & \text{w.p. } \frac{1}{4}-\frac{\tan^{-1}\frac{\omega^2-1}{2\omega}}{2\pi} \\[2mm] \left[\frac{3\omega^2+1}{2\omega}, -1\right]^\top & \text{w.p. } \frac{1}{4}-\frac{\tan^{-1}\frac{\omega^2-1}{2\omega}}{2\pi} \end{cases},$$

$$\text{signGD/Adam:}\quad \boldsymbol{w}_{k,1} = \boldsymbol{w}_{k,0} + a_k\eta\begin{cases} [1,0]^\top & \text{w.p. } 2\frac{\tan^{-1}\frac{\omega^2-1}{2\omega}}{\pi} \\[2mm] [1,1]^\top & \text{w.p. } \frac{1}{2}-\frac{\tan^{-1}\frac{\omega^2-1}{2\omega}}{\pi} \\[2mm] [1,-1]^\top & \text{w.p. } \frac{1}{2}-\frac{\tan^{-1}\frac{\omega^2-1}{2\omega}}{\pi} \end{cases}.$$

**Second iteration.** Next, we use these updates to analyze the second iteration. Tables 8 to 10 include the updates at the second iteration for GD, signGD and Adam, respectively, where we use the conditions on $\omega$ and the (small) initialization scale. Specifically, the small initialization scale helps ensure that the gradient and the corresponding updated neuron are in the same region (in terms of which samples contribute to the gradient for the next iteration). Using the condition on $\omega$, the updates in rows 5 and 7 of Table 8 remain in the direction of the points $\overline{\boldsymbol{z}}_2$ and $\overline{\boldsymbol{z}}_3$, respectively, whereas those in rows 9 and 11 get along the direction of $[1,0]^\top$.

Based on the updates in Table 8, we can write the GD iterate at any time $t > 1$ as:

$$\boldsymbol{w}_{k,t} = \boldsymbol{w}_{k,1} + \frac{\eta\mu(t-1)}{4}\begin{cases} \left[\frac{\omega^2-1}{\omega}, 0\right]^\top & \text{w.p. } \frac{1}{4}+\frac{\tan^{-1}\frac{\omega^2-1}{2\omega}}{2\pi} \\[2mm] -\left[\frac{\omega^2-1}{\omega}, 0\right]^\top & \text{w.p. } \frac{1}{2} \\[2mm] \left[\frac{\omega^2-1}{2\omega}, 1\right]^\top & \text{w.p. } \frac{1}{8}-\frac{\tan^{-1}\frac{\omega^2-1}{2\omega}}{4\pi} \\[2mm] \left[\frac{\omega^2-1}{2\omega}, -1\right]^\top & \text{w.p. } \frac{1}{8}-\frac{\tan^{-1}\frac{\omega^2-1}{2\omega}}{4\pi} \end{cases}.$$

| $4(\boldsymbol{w}_{k,1} - \boldsymbol{w}_{k,0})/(\eta\mu)$ | Prob. | Set $S$ s.t. $\boldsymbol{w}_{k,1}^\top \boldsymbol{x} > 0$ | $4\text{sign}(a_k)\mathbb{E}_{z\sim\mathcal{D}}[\mathbb{1}[\boldsymbol{w}_{k,1}^\top \boldsymbol{x} \geq 0]y\boldsymbol{x}]/\mu$ |
|---|---|---|---|
| $\left[\frac{\omega^2-1}{\omega},0\right]^\top$ | $\frac{\tan^{-1}\frac{\omega^2-1}{2\omega}}{2\pi}$ | $S_2 \cup S_3$ | $\left[\frac{\omega^2-1}{\omega},0\right]^\top$ |
| $-\left[\frac{\omega^2-1}{\omega},0\right]^\top$ | $\frac{\tan^{-1}\frac{\omega^2-1}{2\omega}}{2\pi}$ | $S_1$ | $-\left[\frac{\omega^2+1}{\omega},0\right]^\top$ |
| $\left[\frac{\omega^2+1}{\omega},0\right]^\top$ | $\frac{\tan^{-1}\frac{\omega^2-1}{2\omega}}{2\pi}$ | $S_2 \cup S_3$ | $\left[\frac{\omega^2-1}{\omega},0\right]^\top$ |
| $-\left[\frac{\omega^2+1}{\omega},0\right]^\top$ | $\frac{\tan^{-1}\frac{\omega^2-1}{2\omega}}{2\pi}$ | $S_1$ | $-\left[\frac{\omega^2+1}{\omega},0\right]^\top$ |
| $\left[\frac{\omega^2-1}{2\omega},1\right]$ | $\frac{\frac{\pi}{2}-\tan^{-1}\frac{\omega^2-1}{2\omega}}{4\pi}$ | $S_2$ | $\left[\frac{\omega^2-1}{2\omega},1\right]^\top$ |
| $-\left[\frac{\omega^2-1}{2\omega},1\right]$ | $\frac{\frac{\pi}{2}-\tan^{-1}\frac{\omega^2-1}{2\omega}}{4\pi}$ | $S_1$ | $-\left[\frac{\omega^2+1}{\omega},0\right]^\top$ |
| $\left[\frac{\omega^2-1}{2\omega},-1\right]$ | $\frac{\frac{\pi}{2}-\tan^{-1}\frac{\omega^2-1}{2\omega}}{4\pi}$ | $S_3$ | $\left[\frac{\omega^2-1}{2\omega},-1\right]^\top$ |
| $-\left[\frac{\omega^2-1}{2\omega},-1\right]$ | $\frac{\frac{\pi}{2}-\tan^{-1}\frac{\omega^2-1}{2\omega}}{4\pi}$ | $S_1$ | $-\left[\frac{\omega^2+1}{\omega},0\right]^\top$ |
| $\left[\frac{3\omega^2+1}{2\omega},1\right]^\top$ | $\frac{\frac{\pi}{2}-\tan^{-1}\frac{\omega^2-1}{2\omega}}{4\pi}$ | $S_2 \cup S_3$ | $\left[\frac{\omega^2-1}{\omega},0\right]^\top$ |
| $-\left[\frac{3\omega^2+1}{2\omega},1\right]^\top$ | $\frac{\frac{\pi}{2}-\tan^{-1}\frac{\omega^2-1}{2\omega}}{4\pi}$ | $S_1$ | $-\left[\frac{\omega^2+1}{\omega},0\right]^\top$ |
| $\left[\frac{3\omega^2+1}{2\omega},-1\right]^\top$ | $\frac{\frac{\pi}{2}-\tan^{-1}\frac{\omega^2-1}{2\omega}}{4\pi}$ | $S_2 \cup S_3$ | $\left[\frac{\omega^2-1}{\omega},0\right]^\top$ |
| $-\left[\frac{3\omega^2+1}{2\omega},-1\right]^\top$ | $\frac{\frac{\pi}{2}-\tan^{-1}\frac{\omega^2-1}{2\omega}}{4\pi}$ | $S_1$ | $-\left[\frac{\omega^2+1}{\omega},0\right]^\top$ |

Table 8: Population gradients at the second iteration for GD.

| $\boldsymbol{w}_{k,1} - \boldsymbol{w}_{k,0}$ | Prob. | $4\mathbb{E}_{z\sim\mathcal{D}}[\mathbb{1}[\boldsymbol{w}_{k,1}^\top \boldsymbol{x} \geq 0]y\boldsymbol{x}]/\mu$ | $\boldsymbol{w}_{k,2} - \boldsymbol{w}_{k,1}$ |
|---|---|---|---|
| $\eta[1,0]^\top$ | $\frac{\tan^{-1}\frac{\omega^2-1}{2\omega}}{\pi}$ | $\left[\frac{\omega^2-1}{\omega},0\right]^\top$ | $\eta[1,0]^\top$ |
| $-\eta[1,0]^\top$ | $\frac{\tan^{-1}\frac{\omega^2-1}{2\omega}}{\pi}$ | $\left[\frac{\omega^2+1}{\omega},0\right]^\top$ | $-\eta[1,0]^\top$ |
| $\eta[1,1]^\top$ | $\frac{\frac{\pi}{2}-\tan^{-1}\frac{\omega^2-1}{2\omega}}{2\pi}$ | $\left[\frac{\omega^2-1}{2\omega},1\right]^\top$ | $\eta[1,1]^\top$ |
| $-\eta[1,1]^\top$ | $\frac{\frac{\pi}{2}-\tan^{-1}\frac{\omega^2-1}{2\omega}}{2\pi}$ | $\left[\frac{\omega^2+1}{\omega},0\right]^\top$ | $-\eta[1,0]^\top$ |
| $\eta[1,-1]^\top$ | $\frac{\frac{\pi}{2}-\tan^{-1}\frac{\omega^2-1}{2\omega}}{2\pi}$ | $\left[\frac{\omega^2-1}{2\omega},-1\right]^\top$ | $\eta[1,-1]^\top$ |
| $-\eta[1,-1]^\top$ | $\frac{\frac{\pi}{2}-\tan^{-1}\frac{\omega^2-1}{2\omega}}{2\pi}$ | $\left[\frac{\omega^2+1}{\omega},0\right]^\top$ | $-\eta[1,0]^\top$ |

Table 9: Population gradients at the second iteration for SignGD (Adam, $\beta_1 = \beta_2 = 0$).

Based on the updates in Table 9, we can write the signGD iterate at any time $t$ as:

$$
\boldsymbol{w}_{k,t} = \boldsymbol{w}_{k,0} + \eta t
\begin{cases}
[1,0]^\top & \text{w.p. } \frac{\tan^{-1}\frac{\omega^2-1}{2\omega}}{\pi} \\
-[1,0]^\top & \text{w.p. } \frac{\tan^{-1}\frac{\omega^2-1}{2\omega}}{\pi} \\
[1,1]^\top & \text{w.p. } \frac{1}{4} - \frac{\tan^{-1}\frac{\omega^2-1}{2\omega}}{2\pi} \\
[1,-1]^\top & \text{w.p. } \frac{1}{4} - \frac{\tan^{-1}\frac{\omega^2-1}{2\omega}}{2\pi} \\
-[1,1/t]^\top & \text{w.p. } \frac{1}{4} - \frac{\tan^{-1}\frac{\omega^2-1}{2\omega}}{2\pi} \\
-[1,-1/t]^\top & \text{w.p. } \frac{1}{4} - \frac{\tan^{-1}\frac{\omega^2-1}{2\omega}}{2\pi}
\end{cases}.
$$

| $\boldsymbol{w}_{k,1} - \boldsymbol{w}_{k,0}$ | $4\boldsymbol{g}_{k,0}/(\eta\mu)$ | Prob. | $4\mathbb{E}_{z\sim\mathcal{D}}\big[\mathbb{1}[\boldsymbol{w}_{k,1}^\top\boldsymbol{x}\geq 0]y\boldsymbol{x}\big]/\mu$ | $\boldsymbol{w}_{k,2} - \boldsymbol{w}_{k,1}$ |
|---|---|---|---|---|
| $\eta[1,0]^\top$ | $\big[\frac{\omega^2-1}{\omega},0\big]^\top$ | $\frac{\tan^{-1}\frac{\omega^2-1}{2\omega}}{2\pi}$ | $\big[\frac{\omega^2-1}{\omega},0\big]^\top$ | $\eta[1,0]^\top$ |
| $-\eta[1,0]^\top$ | $-\big[\frac{\omega^2-1}{\omega},0\big]^\top$ | $\frac{\tan^{-1}\frac{\omega^2-1}{2\omega}}{2\pi}$ | $\big[\frac{\omega^2+1}{\omega},0\big]^\top$ | $-\eta\big[\frac{1}{\sqrt{2}}\frac{2\omega^2}{\sqrt{(\omega^2-1)^2+(\omega^2+1)^2}},0\big]^\top$ |
| $\eta[1,0]^\top$ | $\big[\frac{\omega^2+1}{\omega},0\big]^\top$ | $\frac{\tan^{-1}\frac{\omega^2-1}{2\omega}}{2\pi}$ | $\big[\frac{\omega^2-1}{\omega},0\big]^\top$ | $\eta\big[\frac{1}{\sqrt{2}}\frac{2\omega^2}{\sqrt{(\omega^2+1)^2+(\omega^2-1)^2}},0\big]^\top$ |
| $-\eta[1,0]^\top$ | $-\big[\frac{\omega^2+1}{\omega},0\big]^\top$ | $\frac{\tan^{-1}\frac{\omega^2-1}{2\omega}}{2\pi}$ | $\big[\frac{\omega^2+1}{\omega},0\big]^\top$ | $-\eta[1,0]^\top$ |
| $\eta[1,1]^\top$ | $\big[\frac{\omega^2-1}{2\omega},1\big]$ | $\frac{\frac{\pi}{2}-\tan^{-1}\frac{\omega^2-1}{2\omega}}{4\pi}$ | $\big[\frac{\omega^2-1}{2\omega},1\big]^\top$ | $\eta[1,1]^\top$ |
| $-\eta[1,1]^\top$ | $-\big[\frac{\omega^2-1}{2\omega},1\big]$ | $\frac{\frac{\pi}{2}-\tan^{-1}\frac{\omega^2-1}{2\omega}}{4\pi}$ | $\big[\frac{\omega^2+1}{\omega},0\big]^\top$ | $-\frac{\eta}{\sqrt{2}}\big[\frac{(\omega^2-1)/2+(\omega^2+1)}{\sqrt{((\omega^2-1)/2)^2+(\omega^2+1)^2}},1\big]^\top$ |
| $\eta[1,-1]^\top$ | $\big[\frac{\omega^2-1}{2\omega},-1\big]$ | $\frac{\frac{\pi}{2}-\tan^{-1}\frac{\omega^2-1}{2\omega}}{4\pi}$ | $\big[\frac{\omega^2-1}{2\omega},-1\big]^\top$ | $\eta[1,-1]^\top$ |
| $-\eta[1,-1]^\top$ | $-\big[\frac{\omega^2-1}{2\omega},-1\big]$ | $\frac{\frac{\pi}{2}-\tan^{-1}\frac{\omega^2-1}{2\omega}}{4\pi}$ | $\big[\frac{\omega^2+1}{\omega},0\big]^\top$ | $-\frac{\eta}{\sqrt{2}}\big[\frac{(\omega^2-1)/2+(\omega^2+1)}{\sqrt{((\omega^2-1)/2)^2+(\omega^2+1)^2}},-1\big]^\top$ |
| $\eta[1,1]^\top$ | $\big[\frac{3\omega^2+1}{2\omega},1\big]^\top$ | $\frac{\frac{\pi}{2}-\tan^{-1}\frac{\omega^2-1}{2\omega}}{4\pi}$ | $\big[\frac{\omega^2-1}{2\omega},1\big]^\top$ | $\eta\big[\frac{1}{\sqrt{2}}\frac{4\omega^2}{\sqrt{(3\omega^2+1)^2+(\omega^2-1)^2}},1\big]^\top$ |
| $-\eta[1,1]^\top$ | $-\big[\frac{3\omega^2+1}{2\omega},1\big]^\top$ | $\frac{\frac{\pi}{2}-\tan^{-1}\frac{\omega^2-1}{2\omega}}{4\pi}$ | $\big[\frac{\omega^2+1}{\omega},0\big]^\top$ | $-\frac{\eta}{\sqrt{2}}\big[1\frac{2.5\omega^2+1.5}{\sqrt{((3\omega^2+1)/2)^2+(\omega^2+1)^2}},1\big]^\top$ |
| $\eta[1,-1]^\top$ | $\big[\frac{3\omega^2+1}{2\omega},-1\big]^\top$ | $\frac{\frac{\pi}{2}-\tan^{-1}\frac{\omega^2-1}{2\omega}}{4\pi}$ | $\big[\frac{\omega^2-1}{2\omega},-1\big]^\top$ | $\eta\big[\frac{1}{\sqrt{2}}\frac{4\omega^2}{\sqrt{(3\omega^2+1)^2+(\omega^2-1)^2}},-1\big]^\top$ |
| $-\eta[1,-1]^\top$ | $-\big[\frac{3\omega^2+1}{2\omega},-1\big]^\top$ | $\frac{\frac{\pi}{2}-\tan^{-1}\frac{\omega^2-1}{2\omega}}{4\pi}$ | $\big[\frac{\omega^2+1}{\omega},0\big]^\top$ | $-\frac{\eta}{\sqrt{2}}\big[\frac{2.5\omega^2+1.5}{\sqrt{((3\omega^2+1)/2)^2+(\omega^2+1)^2}},-1\big]^\top$ |

Table 10: Population gradients at the second iteration for Adam, $\beta_1 = \beta_2 \approx 1$.

Based on the updates in Table 10, we can write the Adam iterate at any time $t$ as follows:

$$
\boldsymbol{w}_{k,t} = \boldsymbol{w}_{k,0} + \eta\sum_{\tau=1}^{t}
\begin{cases}
[1,0]^\top & \text{w.p. } \frac{\tan^{-1}\frac{\omega^2-1}{2\omega}}{2\pi}\\[2mm]
\frac{1}{\sqrt{\tau}}\Big[-\frac{\omega^2-1+(\tau-1)(\omega^2+1)}{\sqrt{(\omega^2-1)^2+(\tau-1)(\omega^2+1)^2}},0\Big]^\top & \text{w.p. } \frac{\tan^{-1}\frac{\omega^2-1}{2\omega}}{2\pi}\\[2mm]
\frac{1}{\sqrt{\tau}}\Big[\frac{\omega^2+1+(\tau-1)(\omega^2-1)}{\sqrt{(\omega^2+1)^2+(\tau-1)(\omega^2-1)^2}},0\Big]^\top & \text{w.p. } \frac{\tan^{-1}\frac{\omega^2-1}{2\omega}}{2\pi}\\[2mm]
[-1,0]^\top & \text{w.p. } \frac{\tan^{-1}\frac{\omega^2-1}{2\omega}}{2\pi}\\[2mm]
[1,1]^\top & \text{w.p. } \frac{1}{8}-\frac{\tan^{-1}\frac{\omega^2-1}{2\omega}}{4\pi}\\[2mm]
\frac{-1}{\sqrt{\tau}}\Big[\frac{(\omega^2-1)/2+(\tau-1)(\omega^2+1)}{\sqrt{((\omega^2-1)/2)^2+(\tau-1)(\omega^2+1)^2}},1\Big]^\top & \text{w.p. } \frac{1}{8}-\frac{\tan^{-1}\frac{\omega^2-1}{2\omega}}{4\pi}\\[2mm]
[1,-1]^\top & \text{w.p. } \frac{1}{8}-\frac{\tan^{-1}\frac{\omega^2-1}{2\omega}}{4\pi}\\[2mm]
\frac{-1}{\sqrt{\tau}}\Big[\frac{(\omega^2-1)/2+(\tau-1)(\omega^2+1)}{\sqrt{((\omega^2-1)/2)^2+(\tau-1)(\omega^2+1)^2}},-1\Big]^\top & \text{w.p. } \frac{1}{8}-\frac{\tan^{-1}\frac{\omega^2-1}{2\omega}}{4\pi}\\[2mm]
\Big[\frac{1}{\sqrt{\tau}}\frac{3\omega^2+1+(\tau-1)(\omega^2-1)}{\sqrt{(3\omega^2+1)^2+(\tau-1)(\omega^2-1)^2}},1\Big]^\top & \text{w.p. } \frac{1}{8}-\frac{\tan^{-1}\frac{\omega^2-1}{2\omega}}{4\pi}\\[2mm]
\frac{-1}{\sqrt{\tau}}\Big[\frac{(3\omega^2+1)/2+(\tau-1)(\omega^2+1)}{\sqrt{((3\omega^2+1)/2)^2+(\tau-1)(\omega^2+1)^2}},1\Big]^\top & \text{w.p. } \frac{1}{8}-\frac{\tan^{-1}\frac{\omega^2-1}{2\omega}}{4\pi}\\[2mm]
\Big[\frac{1}{\sqrt{\tau}}\frac{3\omega^2+1+(\tau-1)(\omega^2-1)}{\sqrt{(3\omega^2+1)^2+(\tau-1)(\omega^2-1)^2}},-1\Big]^\top & \text{w.p. } \frac{1}{8}-\frac{\tan^{-1}\frac{\omega^2-1}{2\omega}}{4\pi}\\[2mm]
\frac{-1}{\sqrt{\tau}}\Big[\frac{(3\omega^2+1)/2+(\tau-1)(\omega^2+1)}{\sqrt{((3\omega^2+1)/2)^2+(\tau-1)(\omega^2+1)^2}},-1\Big]^\top & \text{w.p. } \frac{1}{8}-\frac{\tan^{-1}\frac{\omega^2-1}{2\omega}}{4\pi}
\end{cases}
$$

$t \to \infty$ **iterations.** Based on the analysis above, we can compute $\lim_{t\to\infty} \frac{\boldsymbol{w}_{k,t}}{t}$ for each algorithm. For GD, we have:

$$\lim_{t\to\infty} \frac{\boldsymbol{w}_{k,t}}{t} = \frac{\eta\mu}{4} \begin{cases} \left[\frac{\omega^2-1}{\omega}, 0\right]^\top & \text{w.p. } \frac{1}{4} + \frac{\tan^{-1}\frac{\omega^2-1}{2\omega}}{2\pi} \\ -\left[\frac{\omega^2-1}{\omega}, 0\right]^\top & \text{w.p. } \frac{1}{2} \\ \left[\frac{\omega^2-1}{2\omega}, 1\right]^\top & \text{w.p. } \frac{1}{8} - \frac{\tan^{-1}\frac{\omega^2-1}{2\omega}}{4\pi} \\ \left[\frac{\omega^2-1}{2\omega}, -1\right]^\top & \text{w.p. } \frac{1}{8} - \frac{\tan^{-1}\frac{\omega^2-1}{2\omega}}{4\pi} \end{cases}.$$

For Adam with $\beta = 0$ or signGD, we have:

$$\lim_{t\to\infty} \frac{\boldsymbol{w}_{k,t}}{t} = \eta \begin{cases} [1,0]^\top & \text{w.p. } \frac{\tan^{-1}\frac{\omega^2-1}{2\omega}}{\pi} \\ [-1,0]^\top & \text{w.p. } \frac{1}{2} \\ [1,1]^\top & \text{w.p. } \frac{1}{4} - \frac{\tan^{-1}\frac{\omega^2-1}{2\omega}}{2\pi} \\ [1,-1]^\top & \text{w.p. } \frac{1}{4} - \frac{\tan^{-1}\frac{\omega^2-1}{2\omega}}{2\pi} \end{cases}.$$

For Adam with $\beta \approx 1$, using the results in Appendix A.3, we have:

$$\lim_{t\to\infty} \frac{\boldsymbol{w}_{k,t}}{t} = \eta \begin{cases} [1,0]^\top & \text{w.p. } \frac{\tan^{-1}\frac{\omega^2-1}{2\omega}}{2\pi} \\ [-m_1,0]^\top & \text{w.p. } \frac{\tan^{-1}\frac{\omega^2-1}{2\omega}}{2\pi} \\ [m_2,0]^\top & \text{w.p. } \frac{\tan^{-1}\frac{\omega^2-1}{2\omega}}{2\pi} \\ [-1,0]^\top & \text{w.p. } \frac{\tan^{-1}\frac{\omega^2-1}{2\omega}}{2\pi} \\ [1,1]^\top & \text{w.p. } \frac{1}{8} - \frac{\tan^{-1}\frac{\omega^2-1}{2\omega}}{4\pi} \\ [1,-1]^\top & \text{w.p. } \frac{1}{8} - \frac{\tan^{-1}\frac{\omega^2-1}{2\omega}}{4\pi} \\ [-m_3,0]^\top & \text{w.p. } \frac{1}{4} - \frac{\tan^{-1}\frac{\omega^2-1}{2\omega}}{2\pi} \\ [m_4,1]^\top & \text{w.p. } \frac{1}{8} - \frac{\tan^{-1}\frac{\omega^2-1}{2\omega}}{4\pi} \\ [m_4,-1]^\top & \text{w.p. } \frac{1}{8} - \frac{\tan^{-1}\frac{\omega^2-1}{2\omega}}{4\pi} \\ [-m_5,0]^\top & \text{w.p. } \frac{1}{4} - \frac{\tan^{-1}\frac{\omega^2-1}{2\omega}}{2\pi} \end{cases},$$

where $m_1, \ldots, m_5$ are constants that satisfy $0.935 \le m_1 \le 1$, $0.923 \le m_2 \le 1$, $0.84 \le m_3 \le 1$, $0.72 \le m_4 \le 1$, $0.98 \le m_5 \le 1$. Taking $s = m_4$ and normalizing each direction then finishes the proof.

$\square$

### A.3 Auxiliary Results

**Lemma 1.** *Given a constant $r > 0$ and function $f_r(x) = \frac{x-1+r}{\sqrt{x(x-1+r^2)}}$, where $x \ge 1$, it holds that $f'_r(x) \ge 0$ when $x \ge 1 + r$. Further, when $x \in \mathbb{N}$, the minima occurs at either $x = 1 + \lfloor r \rfloor$ or $x = 2 + \lfloor r \rfloor$, and it holds that:*

$$\min\left(\frac{\lfloor r \rfloor + r}{\sqrt{(1+\lfloor r \rfloor)(\lfloor r \rfloor + r^2)}}, \frac{1+\lfloor r \rfloor + r}{\sqrt{(2+\lfloor r \rfloor)(1+\lfloor r \rfloor + r^2)}}\right) \le f_r(x) \le 1.$$

The result can be obtained by examining the derivative of $f_r(x)$ with respect to $x$, so we omit the proof.

Further, given $r_1 := \frac{\omega^2-1}{\omega^2+1}$, $r_2 := \frac{\omega^2+1}{\omega^2-1} = 1/r_1$, $r_3 := 0.5r_1$, $r_4 := \frac{3\omega^2+1}{\omega^2-1}$, $r_5 := 0.5\frac{3\omega^2+1}{\omega^2+1}$, and $\omega \geq \frac{1+\sqrt{5}}{2}$, it holds that:

$$0.4472 \leq r_1 < 1, \quad 1 \leq r_2 < 2.236, \quad 0.2236 \leq r_3 < 0.5, \quad 3 \leq r_4 \leq 5.4721, \quad 1.2236 \leq r_5 \leq 1.5.$$

Alternately, for a specific value of $\omega$, we can compute these exactly. For instance, when $\omega = 2$, $r_1 = 0.6$, $r_2 \approx 1.6667$, $r_3 = 0.3$, $r_4 \approx 4.3333$, $r_5 = 1.3$.

Also, we can simplify the lower bound on $f_r(x)$ as follows:

$$c(r) := \min\left( \frac{\lfloor r \rfloor + r}{\sqrt{(1+\lfloor r \rfloor)(\lfloor r \rfloor + r^2)}}, \frac{1 + \lfloor r \rfloor + r}{\sqrt{(2+\lfloor r \rfloor)(1+\lfloor r \rfloor + r^2)}} \right) = \begin{cases} \frac{1+r}{\sqrt{2(1+r^2)}}, & 0 < r < 1 \\ \frac{2+r}{\sqrt{3(2+r^2)}}, & 1 < r < 2 \\ \frac{3+r}{\sqrt{4(3+r^2)}}, & 2 < r < 2\sqrt{3} \\ \frac{4+r}{\sqrt{5(4+r^2)}}, & 2\sqrt{3} < r < 4 \\ \frac{5+r}{\sqrt{6(5+r^2)}}, & 4 < r < 5 \\ \frac{6+r}{\sqrt{7(6+r^2)}}, & 5 < r < 6 \end{cases}.$$

We can use this to obtain the exact lower bounds for the aforementioned intervals:

$$\min_r c(r) \approx \begin{cases} 0.9354, & 0.4472 \leq r < 1, \\ 0.9238, & 1 \leq r < 2.236, \\ 0.8443, & 0.2236 \leq r < 0.5, \\ 0.7240, & 3 \leq r \leq 5.4721, \\ 0.9802, & 1.2236 \leq r \leq 1.5. \end{cases}$$

When $\omega = 2$, $c(r_1) \approx 0.9713$, $c(r_2) \approx 0.9681$, $c(r_3) \approx 0.8803$, $c(r_4) \approx 0.7808$, $c(r_5) \approx 0.9936$.

**Lemma 2.** *The sum* $f(x) := \sum_{\tau=1}^{x} \frac{1}{\sqrt{\tau}}$ *satisfies* $2\sqrt{x} - 2 \leq f(x) \leq 2\sqrt{x} - 1$.

*Proof.* To establish the bounds for $f(x)$, we can compare the sum to the corresponding integral. We have:

$$f(x) = \sum_{\tau=1}^{x} \frac{1}{\sqrt{\tau}} \geq \int_1^{x+1} \frac{1}{\sqrt{n}}\, dn = 2\sqrt{x+1} - 2,$$

$$f(x) = \sum_{\tau=1}^{x} \frac{1}{\sqrt{\tau}} \leq 1 + \int_1^{x} \frac{1}{\sqrt{n}}\, dn = 1 + 2\sqrt{x} - 2 = 2\sqrt{x} - 1.$$

Combining both inequalities and using Using the fact that $\sqrt{x+1} \geq \sqrt{x}$ finishes the proof. $\qquad\square$

# B   Additional Experiments and Details of Settings

All experiments in this section were performed on an internal cluster with NVIDIA V100 and P100 GPUs with 32GB memory each.

We list the licenses under which various datasets used in this work were released as follows. All the datasets are publicly available. MNIST is released under the CC BY-SA 3.0 license. CIFAR-10 is released under the MIT license. The code to generate the Waterbirds dataset is released under the MIT license. The creators of the CelebA dataset encourage its use for non-commercial research purposes only, but do not mention a license name. MultiNLI is released under the ODC-By license. CivilComments is released under the CC BY-NC 4.0 license.

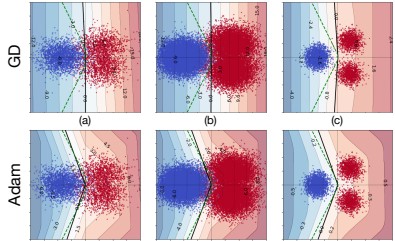

Figure 8: Comparison of the decision boundaries learned by GD and Adam with the Bayes' optimal predictor (dashed green) across three settings of our synthetic setup (details in the text).

## B.1 Gaussian Dataset

**Additional Results.** In this section, we discuss additional results for the Gaussian dataset. Across all settings, we consider full-batch GD with a learning rate of 0.1 and Adam with a learning rate of $10^{-4}$ and $\beta_1 = \beta_2 = 0.9999$. We set $m = 500, \alpha = 0.01, \omega = 2$ for the results in this section. In the finite sample setting, we consider $d = 100$ and $(\sigma_x, \sigma_y, \sigma_z) = (0.2, 0.15, 0.01)$. When $n = 5000$, $\mu = 0.2$, Adam achieves $0.618\%$ better test accuracy than GD. When $n = 30000$, $\mu = 0.25$, the gap is $0.521\%$. In the population setting, we consider $d = 10$, $\mu = 0.3$, $(\sigma_x, \sigma_y, \sigma_z) = (0.1, 0.1, 0.01)$ and Adam achieves $0.292\%$ better test accuracy than GD. The decision boundaries learned by GD and Adam in these three cases are shown in Fig. 8(a), (b), and (c), respectively. In Fig. 3, the test accuracy of Adam is $0.55\%$ more than that of GD.

**Effect of Stochasticity.** We repeated the experiment in Fig. 1 with batch size 50 to see the effect of stochasticity/mini-batch training. The results are shown in Fig. 9. We observe that the decision boundary learned by Adam is more nonlinear than SGD and closer to Bayes' optimal predictor. However, the decision boundary learned by mini-batch Adam is less nonlinear as compared to full-batch Adam.

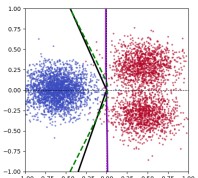

Figure 9: Comparison of the decision boundaries learned by GD and Adam with the Bayes' optimal predictor (dashed green) with batch size 50.

## B.2 MNIST Dataset with Spurious Correlation.

**Training Details.** We train a two-layer NN with width 64 using Adam or SGD with learning rates $10^{-3}$ and 0.1, respectively, using batch size 64. The network is initialized as follows. Hidden layer weights are initialized with small-scale random initialization, specifically from Gaussian distribution with $\sigma = 0.001/\sqrt{d}$, where $d$ is the input dimension. Final layer weights are initialized as $\pm 1/\sqrt{m}$ with equal probability and kept fixed, where $m$ is the hidden dimension. We train using BCE loss till the loss reaches $5e - 3$.

## B.3 Dominoes Dataset

**Dataset.** The Dominoes dataset [9, 12] is composed of images where the top half of the image shows an MNIST digit [13] from class $\{0, 1\}$, while the bottom half shows an image from other image datasets such as MNIST, Fashion-MNIST or CIFAR-10. In our case, we use CIFAR-10 [14] images from classes {automobile, truck}, which is the MNIST-CIFAR dataset. Fig. 1 in Qiu et al. [11] demonstrates how the MNIST-CIFAR dataset is generated as well as more example images.

**Training Details.** We train both a randomly-initialized ResNet-18 and ResNet-34 model for up to 500 epochs or until convergence using a batch size of 32, weight decay $10^{-5}$, initial learning rates $10^{-3}$ for SGD and $10^{-4}$ for Adam, and a cosine annealing learning rate scheduler. We average results across 5 random seeds. The groups in the test set are balanced.

**Additional Results.** Following Kirichenko et al. [24], we generate datasets with spurious correlation strengths of $99\%$ and $95\%$ between the spurious features and the target label, while the core features are fully predictive of the label. We report the final average worst-group accuracies for each correlation strength and optimizer in Table 11 and Table 12 for ResNet-18 and ResNet-34, respectively. In both cases, training with Adam leads to significantly better metrics.

| Method | 99% correlation | | | 95% correlation | | |
| --- | --- | --- | --- | --- | --- | --- |
| | Original Acc. | Core-Only Acc. | Decoded Acc. | Original Acc. | Core-Only Acc. | Decoded Acc. |
| SGD | $0.00_{\pm 0.00}$ | $0.00_{\pm 0.00}$ | $60.42_{\pm 7.06}$ | $0.81_{\pm 0.38}$ | $1.66_{\pm 1.79}$ | $71.04_{\pm 0.63}$ |
| Adam | $0.20_{\pm 0.28}$ | $0.41_{\pm 0.55}$ | $71.37_{\pm 1.67}$ | $14.17_{\pm 3.15}$ | $20.63_{\pm 5.75}$ | $84.66_{\pm 0.18}$ |

Table 11: Worst-group accuracies for original accuracy, core-only accuracy, and decoded accuracy for a ResNet-18 model trained using SGD and Adam.

## B.4 Subgroup Robustness Datasets

**Training Details.** Following Sagawa* et al. [16], for the image datasets, we use the Pytorch `torchvision` implementation of ResNet50 [23] which is pre-trained on the ImageNet dataset, and

| Method | 99% correlation | | | 95% correlation | | |
| --- | --- | --- | --- | --- | --- | --- |
| | Original Acc. | Core-Only Acc. | Decoded Acc. | Original Acc. | Core-Only Acc. | Decoded Acc. |
| SGD | $0.16_{\pm 0.09}$ | $0.00_{\pm 0.00}$ | $47.60_{\pm 8.55}$ | $0.24_{\pm 0.22}$ | $0.00_{\pm 0.00}$ | $59.25_{\pm 3.94}$ |
| Adam | $0.08_{\pm 0.11}$ | $3.32_{\pm 2.97}$ | $69.11_{\pm 3.26}$ | $9.60_{\pm 4.19}$ | $18.31_{\pm 5.32}$ | $82.68_{\pm 1.00}$ |

Table 12: Worst-group accuracies for original accuracy, core-only accuracy, and decoded accuracy for a ResNet-34 model trained using SGD and Adam.

for the language-based datasets, we use the Hugging Face `pytorch-transformers` implementation of the pre-trained BERT `bert-base-uncased` model [22]. We report test results for the epoch/hyperparameter setting with the highest worst-group accuracy on the validation set.

**Main Results.** We use batch size 512 for Waterbirds and MultiNLI and 1024 for CelebA and CivilComments. The results are averaged over five independent runs for the image datasets and four independent runs for the language datasets. We fine-tune until convergence, which takes 7 epochs on the language datasets, 5 epochs on CelebA and 100 epochs on Waterbirds. Fig. 10 shows the hyperparameters (learning rate, weight decay and momentum parameters) considered for the two optimizers across these datasets and the worst-group test accuracies at the last fine-tuning epoch. Fig. 2 and Table 13 show the final results. For Adam, we find that lower values of $\beta_1, \beta_2$ (compared to the default $(0.9, 0.999)$) are generally better: the best values for each of them are $10^{-8}$ or $0.5$, across the four datasets.

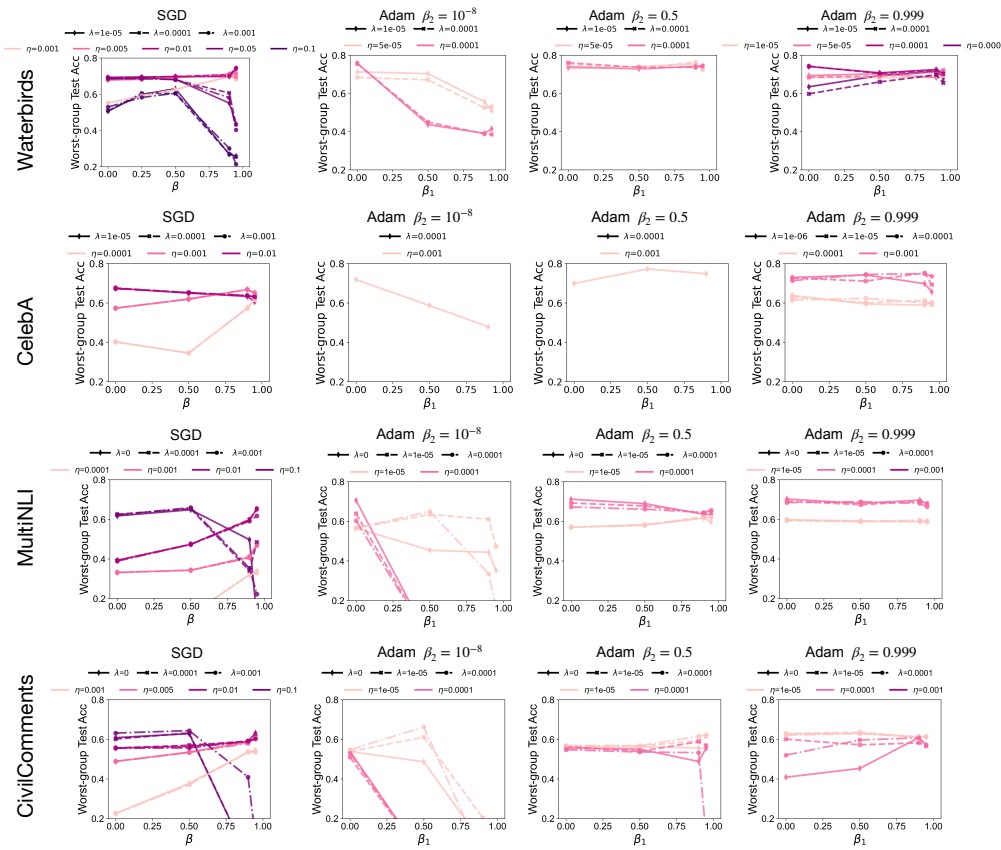

Figure 10: Hyperparameter sweep for SGD and Adam optimizers on four benchmark datasets for subgroup robustness, showing worst-group test accuracy at the last fine-tuning epoch. We sweep the momentum parameters ($\beta$ for SGD, $\beta_1, \beta_2$ for Adam), learning rate ($\eta$), and weight decay ($\lambda$).

**Results with Smaller Batch Sizes.** Here, following Liu et al. [57], we use a batch size of 128 for the image datasets and 32 for the language datasets. The results are averaged over four independent runs, and presented in Table 14. We used the default momentum values for Adam in this setting, but tuned the learning rate and weight decay.

| Optimizer | Waterbirds | | CelebA | | MultiNLI | | CivilComments | |
|---|---|---|---|---|---|---|---|---|
| | Average | Worst-group | Average | Worst-group | Average | Worst-group | Average | Worst-group |
| SGD | $86.9_{\pm1.09}$ | $74.35_{\pm3.26}$ | $93.74_{\pm0.27}$ | $67.56_{\pm1.81}$ | $81.12_{\pm0.22}$ | $65.92_{\pm2.26}$ | $86.68_{\pm0.18}$ | $62.89_{\pm1.24}$ |
| Adam | $87.91_{\pm0.52}$ | $76.04_{\pm2.04}$ | $90.89_{\pm1.70}$ | $77.22_{\pm8.22}$ | $81.99_{\pm0.32}$ | $71.18_{\pm2.74}$ | $85.73_{\pm0.58}$ | $66.25_{\pm3.05}$ |

Table 13: Comparison of average and worst-group test accuracy on four benchmark datasets for subgroup robustness, using larger batch sizes: Adam outperforms SGD. See Section 4 and Appendix B for details.

For the image datasets, we use SGD with momentum $0.9$ and set the learning rate as $10^{-3}$ for Waterbirds and $10^{-4}$ for CelebA and the weight decay as $10^{-4}$ for both datasets. For Adam, tried learning rates $\{10^{-5}, 10^{-4}\}$ and weight decays $\{10^{-6}, 10^{-4}\}$ for both datsets. We use learning rate and weight decay of $10^{-5}$ and $10^{-6}$ for Waterbirds, and $10^{-4}$ and $10^{-4}$ for CelebA for the final results.

For the language datasets, we tried the following settings. For Adam, we tried learning rates $\{10^{-5}, 2 \cdot 10^{-5}\}$ for both datasets. For SGD, we set the learning rate as $10^{-3}$ and tried momentum values $\{0, 0.9\}$. For both optimizers, we tried weight decays $\{0, 10^{-3}\}$. For the final results, we set weight decay as $0$ across all cases and use a learning rate of $10^{-5}$ for Adam for both datasets. The SGD-momentum is set as $0.9$ for MultiNLI and $0$ for CivilComments.

Consistent with the results using larger batch size, we find that even with smaller batch sizes, Adam attains better worst-group test accuracy and comparable average test accuracy compared to SGD, as shown in Table 14. However, we observe that the gains are smaller for image datasets and larger for language datasets in this setting.

| Optimizer | Waterbirds | | CelebA | | MultiNLI | | CivilComments | |
|---|---|---|---|---|---|---|---|---|
| | Average | Worst-group | Average | Worst-group | Average | Worst-group | Average | Worst-group |
| SGD | $85.74_{\pm0.34}$ | $71.95_{\pm0.94}$ | $93.73_{\pm0.32}$ | $67.64_{\pm3.09}$ | $80.17_{\pm0.78}$ | $54.38_{\pm1.59}$ | $72.99_{\pm1.68}$ | $43.72_{\pm4.22}$ |
| Adam | $86.33_{\pm1.09}$ | $73.44_{\pm2.57}$ | $94.10_{\pm1.11}$ | $68.19_{\pm2.66}$ | $81.78_{\pm0.16}$ | $67.21_{\pm1.93}$ | $83.71_{\pm0.98}$ | $69.64_{\pm2.18}$ |

Table 14: Comparison of average and worst-group test accuracy on four benchmark datasets for subgroup robustness, using smaller batch sizes: Adam outperforms SGD. See Section 4 and Appendix B for details.

### B.5 Boolean Features Dataset

**Dataset.** Formally, let $d_c, d_s \in \mathbb{N}$ be the number of core and spurious features respectively, $d_u \in \mathbb{N}$ be the number of independent features that are independent of the label, and let $d := d_c + d_s + d_u$ be the total dimension of the vector. For some $\boldsymbol{x} \in \{-1, +1\}^d$, denote $x_c \in \{-1, +1\}^{d_c}$, $x_s \in \{-1, +1\}^{d_s}$, and $x_u \in \{-1, +1\}^{d_u}$ to be the coordinates of $\boldsymbol{x}$ that correspond to the core, spurious, and independent features respectively. Let $\lambda \in [0, 1]$ be the strength of the spurious correlation. Define two Boolean functions:

$$f_c : \{+1, -1\}^{d_c} \rightarrow \{+1, -1\}, \quad f_s : \{+1, -1\}^{d_s} \rightarrow \{+1, -1\}$$

Next, we define the distributions $\mathcal{D}_{\text{same}}$, $\mathcal{D}_{\text{diff}}$, and $\mathcal{D}_\lambda$. Define $\mathcal{D}_{\text{same}}$ to be the uniform distribution over the set of points in $\{-1, +1\}^d$ where the core and spurious features agree:

$$\mathcal{D}_{\text{same}} := \text{Unif}\left(\left\{\boldsymbol{x} \in \{-1, +1\}^d \ : \ f_c(x_c) = f_s(x_s)\right\}\right)$$

Similarly define $\mathcal{D}_{\text{diff}}$ as the uniform distribution over $\{-1, +1\}^d$ where the core and spurious features disagree:

$$\mathcal{D}_{\text{diff}} := \text{Unif}\left(\left\{\boldsymbol{x} \in \{-1, +1\}^d \ : \ f_c(x_c) \neq f_s(x_s)\right\}\right)$$

Lastly, define $\mathcal{D}_\lambda$ as the distribution where with probability $\lambda$, a sample is drawn from $\mathcal{D}_{\text{same}}$, and otherwise a sample is drawn from $\mathcal{D}_{\text{diff}}$.

**Details of Hyperparameters.** We use a width 20 NN with Leaky ReLU activation with $0.01$ negative slope and train with cross-entropy loss. We use 10000 training samples and 5000 test samples drawn from a data distribution where $d = 50, d_u = 41$, training for 80000 epochs. We consider normal initialization with $\mu = 0$ and $\sigma = \sqrt{\frac{2}{\text{in\_features}}}$. For each optimizer, we did a sweep across various learning rates and picked the best performing learning rate. The results are reported after averaging across 5 random seeds.

**Additional Results.** In Table 15, we consider 4 initialization schemes for the model parameters: (1) the default PyTorch linear initialization, which is a uniform initialization on $(-\sqrt{k}, \sqrt{k})$ where $k = \frac{1}{\text{in\_features}}$, and (2) normal initialization with $\mu = 0$ and $\sigma = \alpha\sqrt{\frac{2}{\text{in\_features}}}$, with $\alpha = 10^{-2}$, $10^{-1}$ & 1. These results are in line with those in Table 4 and show that Adam leads to richer feature learning. We report in Table 16 the final test accuracies, decoded core, and decoded spurious correlations at the end of training for each optimizer, highlighting that Adam maintains its superior performance throughout training. We also report in Table 17 the average test accuracies, decoded core, and decoded spurious correlations at lowest comparable training loss for SGD and Adam, where we use 9 core features instead of 8, so that $d_c = 9$, $d_s = 1$, $d_u = 40$. We use the same correlation strength $\lambda = 0.9$ and uniform initialization.

| Method | Uniform init | | | Normal init ($\alpha = 0.01$) | | | Normal init ($\alpha = 0.1$) | | | Normal init ($\alpha = 1$) | | |
|---|---|---|---|---|---|---|---|---|---|---|---|---|
| | Test acc | DCC | DSC | Test acc | DCC | DSC | Test acc | DCC | DSC | Test acc | DCC | DSC |
| SGD | $95.61_{\pm2.12}$ | $0.75_{\pm0.09}$ | $0.54_{\pm0.15}$ | $95.00_{\pm0.74}$ | $0.75_{\pm0.03}$ | $0.95_{\pm0.04}$ | $96.08_{\pm1.44}$ | $0.79_{\pm0.07}$ | $0.77_{\pm0.20}$ | $89.58_{\pm1.92}$ | $0.51_{\pm0.08}$ | $0.78_{\pm0.08}$ |
| Adam | $\mathbf{97.29}_{\pm0.62}$ | $\mathbf{0.84}_{\pm0.03}$ | $\mathbf{0.42}_{\pm0.07}$ | $\mathbf{98.13}_{\pm0.30}$ | $\mathbf{0.89}_{\pm0.01}$ | $\mathbf{0.40}_{\pm0.02}$ | $\mathbf{97.77}_{\pm0.78}$ | $\mathbf{0.87}_{\pm0.03}$ | $\mathbf{0.42}_{\pm0.08}$ | $\mathbf{97.87}_{\pm0.69}$ | $\mathbf{0.87}_{\pm0.03}$ | $\mathbf{0.36}_{\pm0.06}$ |

Table 15: Test accuracy, decoded core, and decoded spurious correlations averaged across 5 random seeds for SGD and Adam at lowest comparable training loss. Higher test accuracy and decoded core correlation are better. Lower decoded spurious correlation is better.

| Method | Uniform init | | | Normal init ($\alpha = 0.01$) | | | Normal init ($\alpha = 0.1$) | | | Normal init ($\alpha = 1$) | | |
|---|---|---|---|---|---|---|---|---|---|---|---|---|
| | Test acc | DCC | DSC | Test acc | DCC | DSC | Test acc | DCC | DSC | Test acc | DCC | DSC |
| SGD | $95.54_{\pm2.18}$ | $0.75_{\pm0.13}$ | $0.49_{\pm0.20}$ | $94.96_{\pm0.75}$ | $0.75_{\pm0.01}$ | $0.95_{\pm0.04}$ | $96.12_{\pm1.52}$ | $0.81_{\pm0.07}$ | $0.76_{\pm0.22}$ | $89.54_{\pm1.90}$ | $0.51_{\pm0.06}$ | $0.81_{\pm0.10}$ |
| Adam | $\mathbf{99.01}_{\pm0.71}$ | $\mathbf{0.94}_{\pm0.04}$ | $\mathbf{0.31}_{\pm0.09}$ | $\mathbf{99.28}_{\pm0.52}$ | $\mathbf{0.96}_{\pm0.03}$ | $\mathbf{0.30}_{\pm0.07}$ | $\mathbf{99.20}_{\pm0.39}$ | $\mathbf{0.95}_{\pm0.02}$ | $\mathbf{0.36}_{\pm0.03}$ | $\mathbf{99.22}_{\pm0.37}$ | $\mathbf{0.95}_{\pm0.01}$ | $\mathbf{0.20}_{\pm0.08}$ |

Table 16: Final test accuracy, decoded core, and decoded spurious correlations averaged across 5 random seeds for SGD and Adam at the end of training.

**Training Dynamics.** We report in Figures 11 and 12 the test accuracy, decoded core, and decoded spurious correlations throughout training with $d_c = 8$, $d_s = 1$, $\lambda = 0.9$, for 5 different seeds with normal initialization ($\alpha = 1$) for SGD and Adam respectively. From these dynamics,

| Method | Uniform init | | |
|---|---|---|---|
| | Test acc | DCC | DSC |
| SGD | $83.47_{\pm0.94}$ | $0.23_{\pm0.03}$ | $0.74_{\pm0.02}$ |
| Adam | $\mathbf{94.22}_{\pm1.53}$ | $\mathbf{0.70}_{\pm0.06}$ | $\mathbf{0.51}_{\pm0.05}$ |

Table 17: Test accuracy, decoded core, and decoded spurious correlations on the Boolean features dataset for SGD and Adam at lowest comparable training loss. $d_c = 9$, $d_s = 1$, $d_u = 40$.

we observe that SGD learned the spurious feature early in training and gradually learns some of the core feature but still retains the spurious feature information. In contrast, when training with Adam, we see the spurious feature is learned early in training, but forgotten quickly as training progresses to instead learn the core feature.

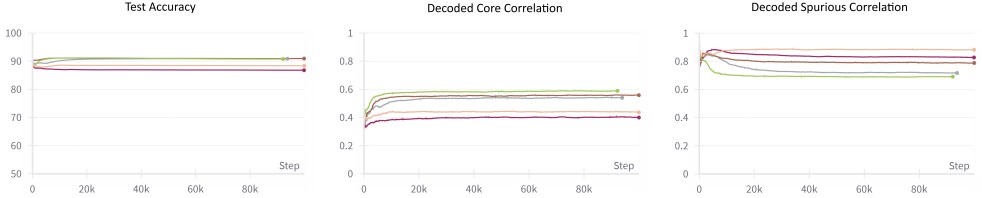

Figure 11: Training curves for SGD with normal weight initialization ($\alpha = 1$) for 5 random seeds.

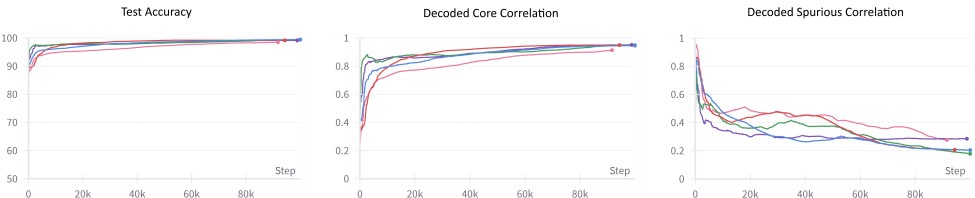

Figure 12: Training curves for Adam with normal weight initialization ($\alpha = 1$) for 5 random seeds.

# C   Discussion

In this section, we discuss some limitations of our work.

First, we note that the theoretical results hold under some assumptions, such as the population setting and with linear loss function, which are quite different from practice. However, given that this is the first result of its kind it is natural to start with some simplifying assumptions. We would like to note that there are no prior works analyzing the training dynamics and implicit bias of signGD or Adam for NNs. The only exception we are aware of is Tsilivis et al. [8], which analyzes steepest descent algorithms (including signGD) for homogeneous NNs trained with an exponentially-tailed loss function. This paper focuses on the late stage of training, and assumes separable data, which does not apply in our setting. Additionally, in their setting (for instance, using uniform disc distributions instead of Gaussian to ensure separability in the population setting), it can be shown that both the linear and the nonlinear predictors are KKT points of the max-$\ell_2$ and max-$\ell_\infty$ margin problems. Consequently, we cannot distinguish between the implicit biases of GD and signGD in their framework. Therefore, it is natural to make some additional assumptions to make the analysis more tractable and take a step towards understanding and contrasting the implicit biases of Adam/signGD and GD. Generalizing these results to broader settings is an interesting direction for future work.

Second, in this work, we focus on settings where simplicity bias in NNs hurts generalization. However, this is not always the case. There is a large body of work showing that simplicity bias in DL is helpful and can explain good in-distribution generalization [58, 59, 60, 61]. Simultaneously, when the goal is to ensure good performance under certain distribution shifts, such as OOD generalization or robustness to spurious features, simplicity bias has been shown to be detrimental in such cases [62, 9, 21]. In this work, we focus on the latter setting to showcase the benefit of richer feature learning encouraged by Adam. However, in general, either one of richer or simpler feature learning may be desirable depending on the problem.

