# OpenReview forum: "The Rich and the Simple: On the Implicit Bias of Adam and SGD"
_NeurIPS.cc/2025/Conference — NeurIPS 2025 poster_

### Official Review · Reviewer_mTkP · 2025-07-01

**Clarity:** 4
**Significance:** 3
**Originality:** 3
**Rating:** 5
**Confidence:** 3

**Summary:**

This paper presents a theoretical and empirical investigation of the simplicity biases of SGD and Adam, giving evidence that SGD can rely on simple, spurious features in a way that may cause reduce out-of-distribution generalization, while Adam tends to learn "better" features. Theoretically, it is proven that when training a two-layer NN (fixed second layer) on a binary classification problem with two-dimensional data coming from a particular gaussian mixture, gradient flow learns an appoximately linear predictor, while signGD learns a non-linear predictor that achieves a lower test loss under certain additional conditions on the data. In a further simplified setting, there are analogous results for GD, signGD, and Adam, although there is no comparison of the test losses in this case. Empirically, it is demonstrated with a large number of synthetic/real datasets that Adam is more robust to spurious correlations than SGD and tends to learn the "core" features instead of spurious features.

**Questions:**

1. Theorem 3 shows conditions under which the test error of signGD is smaller than that of gradient flow. Are there similar conditions under which the opposite conclusion holds?
2. I don't understand the condition on initialization in the statement of Theorem 6, which says that $\|w_{k,0}\| \rightarrow 0$. Can you define this more precisely?  Also, how is this condition used in the proof?
3. How do you reconcile your results with those of Zou et al, who show that GD can generalize better than Adam when training CNNs?
4. In Figure 4, the decision boundaries for Adam and SignGD are super similar. Why is this? I think the paper would be improved by pointing this out and adding some explanation, since in practice we expect Adam to be very different from SignGD.
5. In Table 2, why is agreement with a linear model a good measure of complexity? It seems that you could have low agreement with a baseline linear model while having a very simple decision boundary.  For example, if the baseline linear model has parameter $\theta$, then another linear model with parameter $-\theta$ will have 0 agreement with the baseline linear model.
6. For the experiments in Section 4.2, what were the training loss/accuracy for SGD and Adam? Is it possible that both algorithms have a similar generalization gap but Adam has lower training error?
7. In the setting of Figure 5, did you measure the margins on test set samples? Does it also hold for test data that Adam tends to have larger margins?
8. Do you have any theoretical results that don't use $d=2$? If not, I recommend improving the clarity of the presentation by changing the dataset definition in Equation (1) to remove all dimensions besides two.

**Ethical Concerns:**

["NO or VERY MINOR ethics concerns only"]

**Final Justification:**

I have kept my score the same. The authors have addressed many of my small questions/concerns, and I maintain my positive opinion of the paper, although my confidence is still somewhat low, especially after seeing the proof issues raised by reviewer w3en. I have not thoroughly studied the proofs, and I am getting the impression that there are some details of the theory that could be ironed out. In particular, the comparison of SGD and Adam in Section 3.1 actually compares gradient flow (continuous time) with Adam (discrete time), and it's unclear whether the difference in algorithm behavior could be attributed to this difference rather than the their different learning rules. Also, the full statement of Theorem 2 (Theorem 6 in Appendix C.1) seems a bit informal in how it describes the initialization; I'm not sure whether it is fully rigorous, and the rewording suggested by the authors in their rebuttal does not fix the issue. Still, I appreciate this paper's contribution to our understanding of SGD vs. Adam.

**Limitations:**

Yes

**Quality:**

3

**Strengths And Weaknesses:**

Strengths:
1. The problem considered is very important, and the paper provides a new perspective on the generalization of Adam vs SGD. As pointed out in the paper, it seems a conventional wisdom that SGD generalizes better than Adam, so it is interesting to see evidence of the opposite conclusion. These are the first theoretical results (that I'm aware of) showing a generalization improvement for Adam.
2. The paper contains a lot of empirical results, which gives confidence that the theoretical conclusions are not specific to the synthetic data model. In particular, the experiments with ResNets and CIFAR/MNIST data in Section 4.3 are pretty convincing.

Weaknesses:
1. The data model used for the theory is somewhat artificial and the data is always two-dimensional, which is a big limitation in my eyes. Still, I think it is okay for a first step in this direction.
2. It is not clear whether the complexity of the decision boundary is actually a meaningful way to compare the algorithms. The main conclusion of the theory is that gradient flow learns an approximately linear predictor, while signGD learns a non-linear predictor, and experiments back this up (i.e. Figure 1). But the gap in terms of test accuracy is not as significant, both in theory and practice. Theorem 3 says that signGD's test error is smaller than that of gradient flow under certain conditions, but it is not clear how much smaller or whether we should expect the opposite conclusion to hold under other natural conditions. Similarly, in Figure 1, it is true that Adam's decision boundary is more non-linear than that of SGD, but visually it appears a comparable distance from the Bayes optimal predictor, so is Adam's predictor necessarily better?
3. Theorems 1 and Theorem 2 show a difference between gradient flow and signGD, which is not the same as GD vs signGD. It seems possible to me that the difference in learned features could be due to the continuous time dynamics compared to the discrete time dynamics, and this point is not acknowledged in the paper. I think that the paper will be more transparent if the authors use the name "gradient flow" instead of GD throughout the paper when referring to the results of Theorem 1.

---

> ### Author Rebuttal · Authors · 2025-07-31
>
> Thank you for your time, careful reading and detailed helpful feedback. We are encouraged that you find the problem very important, our perspective on the generalization of Adam vs GD novel and interesting, and appreciate our comprehensive empirical results.
>
> We respond to specific comments below and will incorporate these clarifications in the paper.
>
> **W2: Test accuracy gap for the synthetic setting is not as significant**
>
> We agree that, on the synthetic Gaussian task, absolute accuracy gaps are modest (between 0.2 to 0.6%, depending on the setting). By design, this is a controlled setting to isolate the difference between (S)GD and Adam, and the results serve as corroboration for the theory. The practical impact appears on the larger-scale benchmarks where Adam’s advantage is more significant.
>
>
> **W3, Q8: Presentation**
>
> Thank you for the suggestions, we will incorporate them in the updated version. We will use gradient flow instead of GD to refer to Theorem 1, as suggested. Regarding the dataset definition in Equation (1), our theoretical results are for $d=2$, but we report some empirical results for $d>2$. We will further emphasize that for theory, $d=2$, as suggested.
>
>
> **Q1: Conditions where gradient flow beats signGD?**
>
> Thank you for the question. As discussed in Appendix B, there can be cases where the solution learned by GD is better. For the synthetic setting, the solution learned by gradient flow can have better test accuracy when $\sigma_y$ is much larger than $\sigma_x$. We will add an example in the updated version to demonstrate this concretely. More broadly speaking our goal is not to say one algorithm Adam is better but to provide an explanation why it might be better in some instances. However, there is a kind of no free lunch phenomenon at play here and one can always find settings where one training algorithm is better than another.
>
>
> **Q2: Initialization in Theorem 6**
>
> Thank you for the question. We will state a more precise version of the initialization condition as follows. There exists $\alpha\to 0$ such that $sup_⁡k\|w_{k,0}\|\leq \alpha <<\eta$. Initializing with a small norm ensures that the first and subsequent iterates are dominated by the update directions rather than by the (arbitrary) initial offset, i.e. $w_{k,1}\approx −\eta sign(\nabla L(w_{k,0}))$.
>
>
> **Q3: Reconciling with Zou et al. which shows that Adam generalizes better than SGD for CNNs**
>
> Zou et al. analyze in‑distribution generalization, while the focus of our work is on generalization under distribution shifts. For in-distribution generalization, the simplicity bias exhibited by (S)GD can indeed be beneficial. However, under distribution shifts, such as in the presence of spurious correlations, this simplicity bias can be detrimental and the richer feature learning enabled by Adam can be helpful.
>
> **Q4: Similarity between the predictors learned by Adam vs signGD in the toy setting**
>
> Good observation! In the toy setting, the decision boundary of Adam is slightly more nonlinear than signGD, and the difference is smaller than what we may see in practice. Following the reviewer’s suggestion on highlighting the difference between signGD and Adam, in the updated version of the paper, we will add experimental results for signGD on the group robustness datasets (Section 4.4) to more clearly showcase and discuss the effect of $\beta_1$ and $\beta_2$ on the implicit bias of Adam.
>
>
> **About the MNIST with spurious feature Experiment (Section 4.2)**
>
> **Q6: Train loss/accuracy comparison for SGD and Adam**
>
> For both predictors, the train loss is approximately 0.005 (as mentioned in Appendix D.2), and train accuracy is 99.98%. Thus, Adam has better test accuracy and generalization gap compared to SGD.
>
> **Q5: “Agreement with a linear model” as a complexity proxy**
>
> Thanks for the question. The linear model here serves as the max‑margin linear classifier on the training data (obtained via ERM on the same data, leading to near-perfect train set accuracy). A linear alternative that shows noticeably lower agreement with the max‑margin linear classifier (such as the one suggested by the reviewer) would have a smaller margin or worse accuracy on the train set. In contrast, we see that the NNs trained with SGD or Adam get near-perfect train accuracy. To summarize, a higher capacity model that also attains near‑zero training error, would either have high agreement with this (max-margin) linear classifier and hence, would be close to linear, or have noticeably lower agreement (without sacrificing accuracy), which is achieved by exploiting nonlinear structure.
>
> **Q7: Margin distribution on test set**
>
> Thanks for the suggestion. We checked the margin on the test set, as suggested, and observed a similar trend: Adam has an overall larger margin compared to SGD. To be precise, the mean $\pm$ std dev for the margins across the test set is $0.36\pm 1.22$ for SGD and $1.69\pm 1.75$ for Adam.
>
>
> Thanks a lot for the time and effort to review our work. We appreciate your positive endorsement and support.

---

> > ### Comment · Reviewer_mTkP · 2025-08-05
> >
> > Thanks to the authors for their response. I will keep my score the same, although I still have some thoughts/questions that may help the authors improve the manuscript.
> >
> > 1. Test accuracy gap for the synthetic setting. If anything, I would expect that the controlled setting can amplify the difference between SGD and Adam, so this small gap is still notable to me.
> >
> > 2. Gradient flow vs. gradient descent. I want to emphasize here that the name is not what really matters to me. What really matters is the comparison between gradient flow (continuous time) and Adam (discrete time); this is not an apples-to-apples comparison. How do we know that the difference in their behavior is caused by the different learning rule, as opposed to the fact that one uses discrete time and another continuous time? I strongly recommend that the authors add some discussion along these lines. How do we know that the difference in behavior can really be attributed to SGD vs Adam?
> >
> > 3. Conditions where gradient flow beats signGD. Thank you for acknowledging these cases. I strongly recommend that the authors mention this in the paper, since otherwise the presentation may appear a bit one-sided and incomplete.
> >
> > 4. Initialization in Theorem 6. This condition still does not make sense to me. What do you mean there exists $\alpha \to 0$? Is $\alpha$ fixed or changing? If changing, how is it parameterized? This is also confusing to me because I don't see anywhere in the proof where this initialization is used. What exactly is the condition on the initialization and how is this initialization used in the proof?

---

> > > ### Author Response · Authors · 2025-08-07
> > >
> > > Thank you for your follow-up and helpful suggestions to improve our work. We respond to your questions below and will incorporate all feedback in the paper.
> > >
> > > 1. Gradient flow vs gradient descent: As suggested, we will add a remark to note this difference clearly for our theoretical results. To answer your question about whether the difference in the behavior is caused by the different learning rule, or that one uses discrete time and another continuous time, we note that our numerical simulations in Fig. 1 (as well as other experiments in Section 4.1) corroborate the theoretical results, where we use discrete time updates for both the algorithms. Hence, the difference in the behaviour can be attributed to the different learning rules.
> > >
> > > 2. Initialization in Theorem 6: We will replace the initialization condition $||w_{k,0}||\to 0$ with a fixed bound $\sup_k\||w_{k,0}\||\leq \alpha\leq\eta/2$. The small initialization ensures the first iterate is dominated by the update direction: $w_{k,1}=w_{k,0}-\eta s_0$, where $s_0=sign(\nabla L(w_{k,0}))$, so for each coordinate $i$, $|w_{k,1,i}|\geq \eta-\alpha\geq\eta/2$, hence $w_{k,1}$ is sign-aligned with $-s_0$. Consequently, the next update follows directly from Table 5, and the argument proceeds by that recursion. We will add this clarification in the proof.
> > >
> > >
> > > Thank you again for your engagement and support.

---

### Official Review · Reviewer_w3en · 2025-07-01

**Clarity:** 4
**Significance:** 1
**Originality:** 3
**Rating:** 3
**Confidence:** 5

**Summary:**

A bulk of papers have shown that (S)GD has a bias towards simple solutions in different machine learning tasks. Many view this as a positive property, as it aligns with the Minimum Description Length principle and introduces implicit regularization. This paper comes to undermine this thinking by showing that SGD's solutions are too simple compared to Adam's. The main focus is to demonstrate that the implicit bias of SGD can be counterproductive in some simple settings.

**Questions:**

None

**Ethical Concerns:**

["NO or VERY MINOR ethics concerns only"]

**Final Justification:**

After diving more into the details of the proof, I am now convinced, as I said in my initial review, that this choice of linear loss function is problematic. First, the objective function (risk) is not bounded. This raises the question of how we can measure the effectiveness of the algorithms. Since this is a homogeneous network in a classification task, we can look at convergence in direction, as the authors did. In this regard, we see that gradient flow actually converges in direction to the "optimal" solution. Namely, it does exactly what it is supposed to do. If it doesn't converge to a good separator, it is only because of the bad choice of the objective. In contrast, the adaptive methods don't converge to the optimal direction at all. However, in this carefully designed setting, two wrongs make it right for Adam. Overall, I can't see a real phenomenon here, sorry. This is the main reason why I don't support acceptance.

**Limitations:**

Yes

**Quality:**

2

**Strengths And Weaknesses:**

Strengths:
- Presentation and writing are quite good. The authors did an excellent job of conveying their ideas
- This paper comes to challenge a long-standing convention, and I commend the authors for it.

Weaknesses:
- I am afraid that there might be some flaws in one of the proofs. The proof of Theorem 1 relies on the fact that $ \\frac{d}{dt} \\cos(\\theta\_t ) >0 $ for all $ t \\geq 0 $. From here, the authors conclude that $ \\lim\_{t \\to \\infty} \\cos(\\theta\_t ) = 1 $, which is not true. For example  $ \\theta\_t = \\frac{\\pi}{6}(1+e^{-t}) $ satisfies
$$ \\frac{d}{dt} \\cos(\\theta\_t ) = \\frac{\\pi}{6}e^{-t} \\sin \left(\\frac{\\pi}{6} \left(1+e^{-t} \right)  \right) > 0 \qquad \\forall t \\geq 0$$
but $ \\lim\_{t \\to \\infty} \\cos(\\theta\_t ) = \\frac{\sqrt{3}}{2} $. This can render the Gaussian-Mixture section irrelevant. (Resolved in rebuttal, and the score increased)

- The toy data example is less significant for the following reasons:
1) Both Adam and GD classify 100% correctly for this problem (see Fig. 4), so there is no generalization or other benefit in using Adam.
2) If we use exponential loss instead of linear, then we end up with a separable data+homogeneous network+exponential loss. According to [40], GD will converge to the maximum margin solution, which in this case coincides with the Bayes' optimal predictor (please correct me if I am wrong). So this result depends only on the unusual choice of linear loss function, whereas with the exponential loss, GD has a clear advantage.

Related work:
There are additional papers that you should add to your related work section. For example
- The implicit bias of minima stability: A view from function space
- The Implicit Bias of Minima Stability in Multivariate Shallow ReLU Networks
- On linear stability of SGD and input-smoothness of neural networks

---

> ### Author Rebuttal · Authors · 2025-07-31
>
> Thank you for your time, careful reading and helpful comments to improve our work. We are encouraged by your appreciation of the paper’s writing and presentation, and of challenging a long-standing convention.
>
> We respond to specific comments below and will incorporate these clarifications in the paper.
>
> **W1: Concern about Theorem 1**
>
> Thank you for catching this. We will update the proof of Theorem 1 with the following additional steps that address your concern.
>
> The current proof shows that for neuron $w_k$, $a_k\tfrac{d(cos\theta_{k,t})}{dt} \geq C’ \tfrac{sin^2(\theta_{k,t})}{\||w_{k,t}\||}$
> for some constant $C’>0$.
>
> Using Prop. 1, we can show that the gradients are bounded and consequently, the iterate norm is upper bounded as $\||w_{k,t}\||\leq c(t+1)$, for some constant $c>0$. This gives $a_k\tfrac{d(cos\theta_{k,t})}{dt} \geq C \tfrac{sin^2(\theta_{k,t})}{t+1}$ for some constant $C>0$.
>
> Next, consider $a_k=1$, and suppose $cos⁡\theta_{k,t}​$ stayed below some $L<1$ for all $t$. Then, $sin⁡^2\theta_{k,t}\geq 1−L^2=:m>0$, so $\tfrac{d cos\theta_{k,t}}{dt}\geq\tfrac{Cm}{(1+t)}$. Integrating both sides, we get
> $cos\theta_{k,t}-cos\theta_{k,0}\geq Cm\log(1+t)$,
> which diverges as $t\to\infty$, leading to a contradiction as $|cos(\cdot)|\leq 1$.
>
> Hence, $sin\theta_{k,t}\to 0$, and thus $cos\theta_{k,t} \to sign(a_k)$.
>
>
> **W2: Concern about the toy setting**
>
> Thank you for the thoughtful comments.
> - We agree that both Adam and GD reach 100% training accuracy on this toy distribution. Our goal here is mechanistic: the linear loss lets us compute the population gradients in closed form and see that Adam’s adaptive rescaling uses both features while GD collapses to the dominant (“simpler”) one. This difference shows up in margin: Adam attains a larger margin, so under mild shifts (e.g., additive noise to the first coordinate), Adam generalizes better.
>
> - [40] (Lyu & Li, 2020) proves that for separable, homogeneous networks with exponentially‑tailed loss, GD/flow converges in direction to a KKT point of the parameter‑space $\ell_2$ max‑margin problem, not necessarily the global max‑margin solution. It can be shown that in the toy setup, the linear predictor, the Bayes’ optimal predictor and other predictors with decision boundaries between these predictors, all are KKT points of the $\ell_2$ max-margin problem. Thus, switching to exponential/logistic loss does not necessarily imply GD converges to the Bayes optimal predictor in the toy setup.
>
>
> **W3: Related work**
>
> Thank you for the pointers to related work, we will add these in the updated version of the paper.
>
>
> We hope that these clarifications help address your concerns and that you would consider increasing your score.

---

> > ### Comment · Reviewer_w3en · 2025-08-03
> > **Reply to Rebuttal**
> >
> > Thanks for fixing the proof.
> > Following this fix, I have increased my score.
> >
> > After diving more into the details of the proof, I am now convinced, as I said in my initial review, that this choice of linear loss function is problematic. First, the objective function (risk) is not bounded. This raises the question of how we can measure the effectiveness of the algorithms. Since this is a homogeneous network in a classification task, we can look at convergence in direction, as the authors did. In this regard, we see that gradient flow actually converges in direction to the "optimal" solution. Namely, it does exactly what it is supposed to do. If it doesn't converge to a good separator, it is *only* because of the bad choice of the objective. In contrast, the adaptive methods don't converge to the optimal direction at all. However, in this carefully designed setting, two wrongs make it right for Adam. Overall, I can't see a real phenomenon here, sorry. This is the main reason why I don't support acceptance.
> >
> > It would have been much more interesting and impactful if the authors had shown an advantage in a more conventional setting, for example, in the overparametrized setting for regression, where there exist multiple optimal solutions.

---

> > > ### Author Response · Authors · 2025-08-04
> > >
> > > Thank you for your response and for increasing your score.
> > >
> > > Regarding the choice of the objective, we would like to clarify two points:
> > >
> > > i) **Both gradient flow and signGD reach an optimal solution with linear loss**
> > >
> > > We note that our theoretical results for gradient flow and signGD in Theorems 1 and 2 which use linear loss are for the population setting, where **there are multiple optima**, and both gradient flow and signGD reach an optimal solution. Our main result is that even though both algorithms minimize the loss, the solution learned by signGD is better in terms of test accuracy (both in-distribution and under distribution shifts in certain cases), as shown in Theorem 3 and Section 4.1, and hence closer to the Bayes’ optimal predictor.
> > >
> > > ii) **Same phenomenon is observed in experiments with BCE loss**
> > >
> > > While the theoretical results use a linear loss, our empirical validation of the difference in the implicit bias of SGD and Adam in Section 4.1 includes results with binary cross entropy (BCE) loss. These results show that the phenomenon is **not specific** to the choice of the training objective. Both SGD and Adam minimize the loss but converge to different solutions, and we observe similar neuron trajectories in our experiments as our theoretical results with linear loss. For fair comparison, in all our experiments, we compare the solutions learned by the two algorithms when they achieve comparable train loss.
> > >
> > > The use of linear loss helps simplify the analysis and understand the difference between SGD and Adam that is observed in the experiments with BCE loss. While we agree that it is an important direction to extend the theoretical results to BCE loss, our results serve as a starting point towards understanding this difference.
> > >
> > > Finally, we emphasize that the same phenomenon (SGD exhibits simplicity bias whereas Adam promotes richer feature learning) is observed across various settings beyond the synthetic data, as shown in the extensive experiments on real datasets in Section 4.
> > >
> > > We hope that these clarifications help address your concerns. We are happy to discuss and address any additional questions you may have.

---

> ### Comment · Reviewer_w3en · 2025-08-09
>
> While dealing with an unbounded objective, optima do not exist in the classical sense, only infima. And yes, the linear loss artificially creates additional solutions, since all divergent trajectories are equally good. However, if we look only at the direction, there exists only one optimal direction. In this direction, the derivative of the objective w.r.t. the angle is zero (*i.e.* it is optimal). What you showed is that the direction of gradient flow converges to this direction.
>
> The best way to see that these results are due to this linear loss, is to replace it with an exponential. And as I stated in the initial review, under exponential loss, gradient flow will have the advantage. Additionally, even with linear loss, your proofs are limited to very specific settings (that have some kind of symmetry). If adaptive methods really had an advantage, it should have been easy to generalize your proofs, for example, to separable data. However, I cannot see an easy way to do it.
>
> For clarification, I do believe that adaptive methods have an advantage, but the paper hinges on a problematic setting described above, which depicts the wrong image (the wrong reason why these methods can be superior).

---

> > ### Author Response · Authors · 2025-08-09
> >
> > We thank the reviewer for their response.
> >
> > First, we want to clarify the following for the synthetic setting in our main results in Theorem 1 and 2:
> >
> > **"However, if we look only at the direction, there exists only one optimal direction. In this direction, the derivative of the objective w.r.t. the angle is zero (i.e. it is optimal)."**
> >
> > We respectfully disagree. If we look only at the direction (or the decision boundary of the learned predictor), then the **Bayes’ optimal predictor** is piece-wise linear, **not linear**. We are unsure why the reviewer considers a different notion of optimality (when derivative of the objective w.r.t. the angle is zero). If we assume that the angle refers to the angle of the predictor with the $[1,0,\dots,0]^\top$ direction, then this criterion only considers the set of linear predictors and not nonlinear predictors (which is important to consider since we are training a two-layer ReLU NN). We use 0-1 loss to characterize optimality and we clearly show in the paper that the Bayes’ optimal predictor in this sense is nonlinear and signGD converges to a nonlinear predictor whereas gradient flow converges to a linear predictor (Theorem 1 and 2). Further, the 0-1 loss of signGD is lower than that of gradient flow (Theorem 3).
> >
> > **"The best way to see that these results are due to this linear loss, is to replace it with an exponential. If adaptive methods really had an advantage, it should have been easy to generalize your proofs, for example, to separable data. However, I cannot see an easy way to do it."**
> >
> > We agree with the reviewer that it’s important to ensure that the results are robust to the choice of the loss function. As mentioned in the previous response, our experiments with BCE loss (or logistic loss), which is an **exponentially tailed** loss function, lead to the same conclusion. This includes experiments in overparameterized settings (where the model can interpolate the data) where again we see the same behaviour of Adam vs (S)GD, as discussed in Section 4.1 and Appendix D.1 in the paper. While it would be interesting to extend our proof to this setting for future work, we note that the experimental results clearly demonstrate that the advantage of Adam is robust to the choice of the training objective.
> >
> > Second, if we consider that the reviewer’s concerns are about the **toy setting** in the paper, where the variance is 0, we note that it constitutes only one such setting where we can theoretically show a difference between Adam and gradient descent. We reiterate the main points from our first response regarding the reason we consider this setting:
> >
> > 1) Our goal here is mechanistic: the linear loss lets us compute the population gradients in closed form and see that Adam attains a larger margin, so under mild shifts (e.g., additive noise to the first coordinate), Adam generalizes better.
> >
> > 2) (Lyu & Li, 2020) proves that for separable, homogeneous networks with exponentially‑tailed loss, gradient flow converges in direction to a KKT point of the parameter‑space $\ell_2$ max‑margin problem, not necessarily the global max‑margin solution. So gradient flow does not necessarily have an “advantage”.
> >
> > We hope these clarifications address the reviewer’s concerns and they will consider increasing their score.

---

> > > ### Comment · Reviewer_w3en · 2025-08-09
> > >
> > > Kindly note that my statement about "the derivative of the objective w.r.t. the angle is zero (*i.e.*, it is optimal)" refers to the optimal direction of the optimized objective, which determines the solution you get for the different algorithms, and not Bayes’ optimal predictor.
> > >
> > > Thanks for your replies. I have already increased my score, and I won't increase it further due to the issues I described above.

---

### Official Review · Reviewer_uF13 · 2025-07-02

**Clarity:** 3
**Significance:** 4
**Originality:** 4
**Rating:** 5
**Confidence:** 3

**Summary:**

The paper compares the implicit bias of Adam to that of Gradient Descent (GD) and shows that in some settings GD converges to "simple" solution, whereas Adam is able to capture richer features. The paper contains theoretical results as well as a range of experimental results. While the theoretical results are derived for one specific type of input and for a simple model, the experimental results explore a number of different datasets and models.

**Questions:**

See above.
* Can you give a longer more extensive intuitive explanation for the effects observed? Or is this difficult
* Does Theorem 1 hold regardless of initialization?

**Ethical Concerns:**

["NO or VERY MINOR ethics concerns only"]

**Final Justification:**

I have taken note of some concerns raised in another reviews regarding the loss function used and, just in general, the setting that is studied formally here is very specialized. However, for me the inclusion of the experiments partly make up for those weaknesses and so I would like to keep my positive score.

**Limitations:**

Yes.

**Paper Formatting Concerns:**

No.

**Quality:**

3

**Strengths And Weaknesses:**

Strengths:
* The implicit bias of Adam is far less well studied than the implicit bias of GD. So it is great that this paper adds to the investigation of Adam.
* The experiments are done using simple, but insightful, clean settings. They are also relatively broad and add coherently to the story.
* While the derivations of some of the theoretical results are a bit brute force (which may be necessary, I am not claiming to know a simpler way), I found the paper quite well written in general.

Weakness:
* As common for these kinds of things, the theoretical results are quite special. The model is a two-layer ReLU network, but the second layer weights are not even trainable.
* There could be even more of an intuitive explanation or intuition why Adam behaves the way it does.
* Pleas provide code. While I agree that it seems that the code should be pretty standard, it would allow for much easier replication. Also, certain details are difficult to find in the paper. For example, for the MNIST Dataset with Spurious Features, you say you train a two-layer neural network, but what is the width?

---

> ### Author Rebuttal · Authors · 2025-07-31
>
> Thank you for your time, careful reading and helpful feedback. We are encouraged that you appreciate the paper’s contribution to the study of implicit bias of Adam, and find the experimental settings simple yet insightful, clean and broad, and the paper well-written overall.
>
> We respond to specific comments below and will incorporate these clarifications in the paper.
>
> **W2, Q1: Intuition on Adam’s behaviour vs GD**
>
> Thanks for the question. In our setting, different latent features have different effective scales. GD treats all coordinates uniformly, so the largest‑margin (“simplest”) feature dominates. Adam adaptively rescales coordinates by their second‑moment estimates, approximately normalizing these scale differences; this allows smaller / higher‑frequency features to be learned rather than being suppressed by the dominant one.
>
>
> **W3: Regarding Section 4.2 and Code**
>
> We used MLP width 64 for the experiment in Section 4.2 (MNIST dataset with spurious feature). As we cannot share external links during the rebuttal, we will release the code for the experiment upon acceptance, as suggested.
>
> **W1: Fixed second layer weights in the analysis of two-layer ReLU NN**
>
> We fix the second‑layer weights to isolate and make analyzable the hidden‑layer dynamics. This is a standard practice, e.g., [1-4].
>
> **Q2: Does Theorem 1 hold for any initialization?**
>
> Yes, Theorem 1 holds for any random initialization of the hidden weights ($w_k$). The only condition is that the output weights $a_k$​ are symmetric ($a_k=\pm 1$ w.p. 0.5).
>
>
> Thanks a lot for the time and effort to review our work. We appreciate your positive endorsement and support.
>
>
> **References:**
>
> [1] Frei et al., “Benign Overfitting without Linearity: Neural Network Classifiers Trained by Gradient Descent for Noisy Linear Data,” COLT 2022.
>
> [2] Cao et al., “Benign Overfitting in Two-layer Convolutional Neural Networks,” NeurIPS 2022.
>
> [3] Frei et al., “Implicit Bias in Leaky ReLU Networks Trained on High-Dimensional Data,” ICLR 2023.
>
> [4] Kou et al., “Implicit Bias of Gradient Descent for Two-layer ReLU and Leaky ReLU Networks on Nearly-Orthogonal Data,” NeurIPS 2023.

---

> > ### Comment · Reviewer_uF13 · 2025-08-04
> >
> > Thank you for your response.
> >
> > I agree that fixing second layer weights is not very unusual, in particular in works that study training dynamics/implicit bias in a rigorous theoretical way. At the same time, there are also works that demonstrate that this can make a real difference on the training. The context is very different, but, for example, Goldt et al. [5] demonstrate what the potential impact of fixing the second layer can have in certain settings. So I am not entirely sure if or in what way this aspect of the setup impacts to what extend the general message of your paper can be assumed to generalize. It may deserve a bit of a warning in the conclusions emphasizing this aspect of the setup.
> >
> > Nevertheless, I think the provided experiments using ResNet architectures partly make up for this and therefore I am happy to confirm my positive rating.
> >
> > [5] Goldt et al., "Dynamics of stochastic gradient descent for two-layer neural networks in the teacher-student setup," NeurIPS 2019.

---

> > > ### Author Response · Authors · 2025-08-07
> > >
> > > Thank you for your response. As suggested, we will include a remark on the potential impact of fixing vs training the second layer of the model in some cases. We appreciate your helpful feedback and support.

---

### Official Review · Reviewer_xUiG · 2025-07-05

**Clarity:** 3
**Significance:** 3
**Originality:** 3
**Rating:** 4
**Confidence:** 3

**Summary:**

This paper demonstrates that, in comparison to Adam, stochastic gradient descent exhibits a simplicity bias when training two-layer ReLU neural networks  on a binary classification task with Gaussian data. The empirical analysis also highlights how SGD preferentially converges to simpler models.

**Questions:**

I did not thoroughly examine the proof, but a critical question arises: does the conclusion hold for all two-layer ReLU neural networks on a binary classification task with Gaussian data, or only under specific conditions such as particular final weight configurations and neuron initializations? This distinction is crucial because it determines whether the observed simplicity bias is an isolated phenomenon or a universal property of such models. Clarifying this would strengthen the paper’s contribution by addressing whether the result is broadly applicable or contingent on specific architectural or initialization choices. I will change my score according to the authors response.

**Ethical Concerns:**

["NO or VERY MINOR ethics concerns only"]

**Final Justification:**

The authors have explained about the limitation and I think it's a interesting work.

**Limitations:**

see questions

**Quality:**

3

**Strengths And Weaknesses:**

Strengths:

The paper introduces a compelling theoretical framework by leveraging the Bayes optimal predictor as a benchmark since there are no prior works analyzing the training dynamics and implicit bias of signGD or Adam for NNs. The derivation of signGD as a first step in the analysis is particularly novel and insightful.

Weaknesses:

The theoretical guarantees seems to rely on assumptions about initialization and final weights. The analysis of Adam with momentum remains incomplete (but it's still a good start)

---

> ### Author Rebuttal · Authors · 2025-07-31
>
> Thank you for your time and helpful feedback. We are encouraged that you find our theoretical framework to contrast the implicit bias of Adam and GD in NNs compelling, and the analysis novel and insightful.
>
> **Q: Do the results hold under specific conditions such as particular final weight configurations and neuron initializations?**
>
> Thanks for the question about the scope of our theoretical results.
>
> Theorem 1 and Theorem 2 hold for any random (small scale) initialization of the hidden weights ($w_k$). There is no constraint on the directions in which the hidden weights are initialized. This is in accordance with standard initialization conditions used in the deep learning theory literature as well as the Pytorch default Kaiming initialization (upto a constant). The only condition is that the output weights $a_k$​ are symmetric ($a_k=\pm 1$ w.p. 0.5).
>
> Importantly, we **do not** assume any condition on the final weight configuration. The limiting directions (as stated in the final equations in Theorems 1 and 2) arise from the training dynamics themselves. These equations present the limiting directions each neuron converges to, depending on how it was (randomly) initialized. We also show how the neurons evolve over time in practice in Fig. 3 (second row), which matches the theoretical results. We will further clarify this in the paper.
>
> We hope that this clarification helps address your concern and that you would consider increasing your score.

---

> > ### Comment · Reviewer_xUiG · 2025-08-05
> >
> > Thank you for your detailed clarification regarding the initialization and convergence conditions of your theoretical results. Your response directly addresses my concern about the generality of the assumptions.  In light of this explanation, I have updated my score.

---

> > > ### Author Response · Authors · 2025-08-07
> > >
> > > Thank you for your helpful feedback and for your positive assessment and support.

---

### Decision · Program_Chairs · 2025-09-17

**Decision:**

Accept (poster)

**Comment:**

This paper provides a theoretical and empirical comparison of the implicit biases of SGD and Adam, particularly in settings with structured data and linear losses. The authors demonstrate that Adam tends to prefer nonlinear predictors that approximte Bayes' rule, while SGD prefers linear predictors, offering valuable insights into how optimization algorithms affect generalization.

Even though the setting is rather restrictive (e.g., linear loss, structured data distribution), this work clearly illustrates the distinct implicit biases of GD and Adam, which is an important contribution. The analysis is rigorous and well-structured, and the observations are novel and relevant to the broader understanding of optimization in deep learning.